# Momentum Diminishes the Effect of Spectral Bias in Physics-Informed Neural Networks

## Abstract

Physics-informed neural network (PINN) algorithms have shown promising results in solving a wide range of problems involving partial differential equations (PDEs). However, even the simplest PDEs, often fail to converge to desirable solutions when the target function contains high-frequency modes, due to a phenomenon known as spectral bias. In the present work, we exploit neural tangent kernels (NTKs) to investigate the training dynamics of PINNs evolving under stochastic gradient descent with momentum (GDM). This demonstrates GDM significantly reduces the effect of spectral bias. We have also examined why training a model via the Adam optimizer can accelerate the convergence while reducing the spectral bias. Moreover, our numerical experiments have confirmed that wide-enough networks using GDM or Adam still converge to desirable solutions, even in the presence of high-frequency features.

## 1 Introduction

Physics-informed neural networks (PINNs) have been proposed as alternatives to traditional numerical partial differential equations (PDEs) solvers (Raissi et al., 2019; 2020; Sirignano & Spiliopoulos, 2018; Tripathy & Bilionis, 2018). In PINNs, a PDE which describes the physical domain knowledge of a problem is added as a regularization term to an empirical loss function. Although PINNs has shown remarkable performance in solving a wide range of problems in science and engineering (Cai et al., 2022; Kharazmi et al., 2019; Sun et al., 2020; Kissas et al., 2020; Tartakovsky et al., 2018), regardless of the simplicity of a PDE itself, they often fail to converge to accurate solutions when the target function contains *high-frequency* features (Krishnapriyan et al., 2021; Wang et al., 2021). This phenomenon known as the *spectral bias* exists in even the simplest linear PDEs (Wang et al., 2021; Moseley et al., 2021; Krishnapriyan et al., 2021).

Spectral bias is not limited to PINNs. Rahaman et al. (2019) empirically showed that all fully-connected feed-forward neural networks (NNs) are biased against learning complex components of target functions. Furthermore, Cao et al. (2019) theoretically proved that in training infinitely-wide networks with squared loss, the corresponding eigenvalues of the neural tangent kernel (NTK) (Jacot et al., 2018) indicate the exact convergence rate for different components of the target functions. Thus, spectral bias happens when the absolute values of some of the eigenvalues of the NTK are large while others are small. Recently, utilizing the NTK of infinitely-wide PINNs, Wang et al. (2022) examined the gradient flow of these networks during training. They proved that the training error decays based on $e^{-\kappa_i t}$, where $\kappa_i$ are the eigenvalues of the NTK. Thus, the components of the target function corresponding to the smaller eigenvalues have a slower rate of decay, which causes spectral bias. To tackle the issue of spectral bias, they proposed to assign a weight to each term of the loss function and dynamically update it. Although the results showed some improvements, as the frequency of the target function increased, their proposed PINN still failed to converge to solutions of PDEs. Moreover, as assigning weights can result in indefinite kernels, the training process could become extremely unstable. Of note, compared to the typical NNs, analyzing the effect of spectral bias for PINNs is more challenging as the loss function is regularized by means of adding the PDE equation. Thus, Wang et al. (2022)'s study was limited to training the model only based on GD.

Some studies proposed an alternative approach in which instead of modifying the loss function terms, a high-frequency PDE is solved in a few successive steps. In these methods, it is assumed that the optimal solution of low-frequency PDEs is close to the optimal solution of high-frequency

PDEs. Hence, instead of randomly initializing weights they are being initialized using the optimal solution of low-frequency PDEs. Moseley et al. (2021) implemented a finite element approach where PINNs were trained to learn basis functions over several small, overlapping subdomains. Similarly, Krishnapriyan et al. (2021) proposed a learning method based on learning the solution over small successive chunks of time. Moreover, they proposed another sequential learning scheme in which the model was gradually trained on target functions with lower frequencies, and, at each step, the optimized weights were used as the warm initialization for higher-frequency target functions. In a similar approach, Huang & Alkhalifah (2021) proposed to use the pre-trained models from low-frequency functions and to increase the size of the network (neuron splitting) as the frequency of the target function is increased. Although these methods showed good performance on some PDEs, as the frequency terms became larger, the process became much slower as the required time steps would significantly grow.

In this work, we study the spectral bias of PINNs from an optimization perspective. Existing studies only have focused on effect of the vanilla GD (Wang et al., 2022) or they are limited to some weak empirical evidence indicating that Adam might learn high feature faster (Taylor et al., 2022). We prove that an infinitely-wide PINN under the vanilla GD optimization process will converge to the solution. However, for high-frequency modes, the learning rate needs to become very small, which makes the convergence extremely slow, and hence not possible in practice. Moreover, we prove that for infinitely-wide networks, using the GD with momentum (GDM) optimizer can reduce the effect of spectral bias in the networks, while significantly outperforming vanilla GD. We also investigate why the Adam optimizer can also accelerate the optimization process while decreases the effect of spectral bias in PINNs. To the best of our knowledge this is the first time that the gradient flow of the output of PINNs under the GDM, and Adam are being analyzed, and their relation to solving spectral bias is discussed. Finally, our extensive numerical experiments on sufficiently wide networks confirm our theoretical findings.

## 2 PRELIMINARIES

### 2.1 PINNS GENERAL FORM

The general form of a well-posed PDE on a bounded domain ($\Omega \subset \mathbf{R}^d$) is defined as:

$$\begin{aligned} \mathcal{D}[u](\boldsymbol{x}) = f(\boldsymbol{x}), & \quad \boldsymbol{x} \in \Omega \\ u(\boldsymbol{x}) = g(\boldsymbol{x}), & \quad \boldsymbol{x} \in \partial\Omega \end{aligned} \tag{1}$$

where $\mathcal{D}$ is a differential operator and $u(\boldsymbol{x})$ is the solution (of the PDE), in which $\boldsymbol{x} = (x_1, x_2, \ldots, x_d)$. Note that for time-dependent equations, $\boldsymbol{t} = (t_1, t_2, \ldots, t_d)$ are viewed as additional coordinates within $\boldsymbol{x}$. Hence, the initial condition is viewed as a special type of Dirichlet boundary condition and included in the second term.

Using PINNs, the solution of Eq. 1 can be approximated as $u(\boldsymbol{x}, \boldsymbol{w})$ by minimizing the following loss function:

$$\mathcal{L}(\boldsymbol{w}) := \underbrace{\frac{1}{N_b} \sum_{i=1}^{N_b} \left( u(\boldsymbol{x}_b^i, \boldsymbol{w}) - g(\boldsymbol{x}_b^i) \right)^2}_{\mathcal{L}_b(\boldsymbol{w})} + \underbrace{\frac{1}{N_r} \sum_{i=1}^{N_r} \left( \mathcal{D}[u](\boldsymbol{x}_r^i, \boldsymbol{w}) - f(\boldsymbol{x}_r^i) \right)^2}_{\mathcal{L}_r(\boldsymbol{w})} \tag{2}$$

where $\{\boldsymbol{x}_b^i\}_{i=1}^{N_b}$ and $\{\boldsymbol{x}_r^i\}_{i=1}^{N_r}$ are boundary and collocation points respectively, and $\boldsymbol{w}$ describes the neural network parameters. $\mathcal{L}_b(\boldsymbol{w})$ corresponds to the mean squared error of the boundary (and initial) condition data points, and $\mathcal{L}_r(\boldsymbol{w})$ encapsulates the physics of the problem using the randomly selected collocation points. Similar to all other NNs, minimizing the loss function $\mathcal{L}(\boldsymbol{w})$ results in finding the optimal solutions $\boldsymbol{w}^*$.

### 2.2 INFINITELY-WIDE NEURAL NETWORKS

A fully-connected infinitely-wide NN with $L$ hidden layers can be written as Jacot et al. (2018):

$$\begin{aligned} u_h(\boldsymbol{x}) &= \frac{1}{N} \boldsymbol{\Theta}_h \cdot \boldsymbol{x}_h + \mathbf{b}_h \\ \boldsymbol{x}_{h+1} &= \sigma(u_h) \end{aligned}$$

where $\boldsymbol{\Theta}_h$ and $\mathbf{b}_h$ are respectively the weight matrices and the bias vectors in the layers $h = 1, \ldots, L$, $N$ is the width of the layer, and $\sigma(\cdot)$ is a $\beta$-smooth activation function (e.g. $\tanh(\cdot)$). The final output of the NN is written as:

$$u(\boldsymbol{x}, \boldsymbol{w}) = \frac{1}{N}\boldsymbol{\Theta}_L \cdot \boldsymbol{x}_L + \mathbf{b}_L$$

where $\boldsymbol{w} = (\boldsymbol{\Theta}_0, \mathbf{b}_0, \ldots, \boldsymbol{\Theta}_L, \mathbf{b}_L)$. At each time step $t$, we can determine the change of the output with respect to the input, which defines a NTK:

$$\boldsymbol{K}^t_{(\boldsymbol{x}, \boldsymbol{x}')} = \nabla_w u(\boldsymbol{x}, \boldsymbol{w}(\boldsymbol{t}))^\top \nabla_w u(\boldsymbol{x}', \boldsymbol{w}(\boldsymbol{t})).$$

It is worth noting that the NTK is associated with the model and is completely independent of the choice of optimization algorithms or the loss function. Liu et al. (2020) showed that for a fully-connected NN with a linear output layer, the spectral norm of its Hessian satisfies $\|\mathcal{H}(\boldsymbol{w})\| = \mathcal{O}(\frac{1}{\sqrt{N}})$. Consequently, as the width become larger, the norm of the Hessian becomes smaller: $\lim_{N \to 0} \|\mathcal{H}(\boldsymbol{w})\| = 0$. Thus, one major consequence of dealing with infinitely-wide fully-connected NNs is that if the last layer of the network is linear, the NTK becomes static (not changing over iterations), and the output of the network can be linearized (Liu et al., 2020):

$$u_t^{\text{lin}}(\boldsymbol{w}) \approx u(\boldsymbol{w})|_{\boldsymbol{w}_0} + (\boldsymbol{w} - \boldsymbol{w}_0)\nabla u(\boldsymbol{w})|_{\boldsymbol{w}_0}.$$

# 3 THEORETICAL RESULTS

## 3.1 CONVERGENCE OF GRADIENT DESCENT IN THE PRESENCE OF HIGH-FREQUENCY FEATURES

Generally, the optimization problems corresponding to over-parametrized systems, even on a local scale, are non-convex (Liu et al., 2022). A loss function $\mathcal{L}(\boldsymbol{w})$ of a $\mu$-uniformly conditioned NN satisfies the $\mu$-PL$_*$ condition on a set $S \subset \mathbf{R}^m$ if:

$$\|\nabla_w(\mathcal{L}(\boldsymbol{w}))\|^2 \geq \mu\mathcal{L}(\boldsymbol{w}) \text{ for all } \boldsymbol{w} \in S \tag{3}$$

where $\mu$ is the lower bound of the tangent kernel $\boldsymbol{K}(\boldsymbol{w})$ of the NN. It has been shown that infinitely-wide networks satisfy the $\mu$-PL$_*$ condition, a variant of the Polyak-Lojasiewicz condition, and as a result the (stochastic) gradient descent (SGD/GD) optimization algorithms will converge to the optimal solution (Liu et al., 2022). The following proposition makes use of the $\mu$-PL$_*$ condition to provide a convergence analysis for an infinitely-wide PINN with a loss function as in Eq. 2 optimized with GD (Appendix A).

**Proposition 1.** *Let $\lambda_{b_{\max}}$ and $\lambda_{r_{\max}}$, respectively, be the largest eigenvalues of the Hessians $\nabla^2 \mathcal{L}_b(\boldsymbol{w}^t)$ and $\nabla^2 \mathcal{L}_r(\boldsymbol{w}^t)$. Consider an infinitely-wide PINN optimized with the following update rule:*

$$\boldsymbol{w}_{t+1} = \boldsymbol{w}_t - \eta\nabla\mathcal{L}(\boldsymbol{w}_t),$$

*where $\eta$ is a constant learning rate. Then, provided $\eta = \mathcal{O}(1/(\lambda_{b_{\max}} + \lambda_{r_{\max}}))$, the $\mu$-PL$_*$ condition is satisfied and the PINN will converge.*

Although Proposition 1 provides a convergence guarantee for an infinitely wide-wide PINN, for PDEs exhibiting stiff dynamics (those with high frequency modes), the eigenvalues (of the Hessian) dictating convergence are often very large (Wang et al., 2021). For example, take the one-dimensional Poisson equation:

$$\frac{\partial^2 u}{\partial x^2} = f(x), x \in \Omega \tag{4}$$
$$u(x) = g(x), x \in \partial\Omega$$

with $u(x) = \sin(Cx)$. The norm of the Hessian $\mathcal{H}(\boldsymbol{w}_t)$ is of order $\mathcal{O}(C^4)$ (Appendix B). Thus, as $C$ grows so does the bound on the eigenvalues of $\mathcal{H}(\boldsymbol{w}_t)$, indicating that at least one of $\lambda_{b_{\max}}$ or $\lambda_{r_{\max}}$ will be large. Consequently, the learning rate must be prohibitively small to guarantee convergence. Therefore GD is unusable in practice.

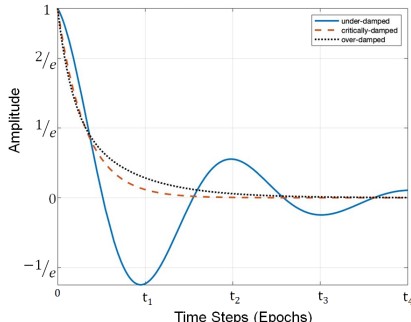

Figure 1: Damped harmonic oscillators show three different characteristics.

## 3.2 WIDE PINN OPTIMIZATION USING GD WITH MOMENTUM

In order to accelerate learning, we investigate the training dynamics of PINNs under GD with momentum (GDM) (Du, 2019):

$$\boldsymbol{w}_{t+1} = \boldsymbol{w}_t + \alpha(\boldsymbol{w}_t - \boldsymbol{w}_{t-1}) - \eta \nabla_w \mathcal{L}(\boldsymbol{w}), \tag{5}$$

where $\alpha$ and $\eta$ are fixed rates. Specifically, we analyze the gradient flow of the update rule in Eq. 5 by leveraging the notion of the NTK. The following theorem (proof in Appendix C) reveals that GDM has a different convergence behavior from GD:

**Theorem 1.** *For an infinitely-wide PINN, the gradient flow of GDM is:*

$$m \begin{bmatrix} \ddot{u}(\boldsymbol{x}_b, \boldsymbol{w(t)}) \\ D[\ddot{u}](\boldsymbol{x}_r, \boldsymbol{w(t)}) \end{bmatrix} = -\mu \begin{bmatrix} \dot{u}(\boldsymbol{x}_b, \boldsymbol{w(t)}) \\ D[\dot{u}](\boldsymbol{x}_r, \boldsymbol{w(t)}) \end{bmatrix} - \boldsymbol{K} \begin{bmatrix} u(\boldsymbol{x}_b, \boldsymbol{w(t)}) - g(\boldsymbol{x}_b) \\ D[u](\boldsymbol{x}_r, \boldsymbol{w(t)}) - h(\boldsymbol{x}_r) \end{bmatrix}$$

*that is analogous to a point mass $m$ undergoing a damped harmonic oscillation in a viscous medium with a friction coefficient of $\mu(\alpha)$ that is function of $\alpha$, Furthermore, $\boldsymbol{K}$ is defined as:*

$$\boldsymbol{K} = \begin{bmatrix} \boldsymbol{K}_{bb} & \boldsymbol{K}_{rb} \\ \boldsymbol{K}_{br} & \boldsymbol{K}_{rr} \end{bmatrix},$$

*where:*

$$\boldsymbol{K}_{bb_{(x,x')}} = \nabla_w u(\boldsymbol{w}, \boldsymbol{x})^\top \nabla_w u(\boldsymbol{w}, \boldsymbol{x}')$$

$$\boldsymbol{K}_{br_{(x,x')}} = \nabla_w u(\boldsymbol{w}, \boldsymbol{x})^\top \nabla_w D[u](\boldsymbol{w}, \boldsymbol{x}')$$

$$\boldsymbol{K}_{rr_{(x,x')}} = \nabla_w D[u](\boldsymbol{w}, \boldsymbol{x}')^\top \nabla_w D[u](\boldsymbol{w}, \boldsymbol{x}')$$

*are three NTKs associated with the boundary and residual terms. Moreover, let $\gamma = \mu/2m$, $\kappa_i$ be the $i$-th eigenvalue of $\boldsymbol{K}$, and $\kappa'_i = \frac{\kappa_i}{m}$. Then, the solutions to the gradient flow are of the form:*

$$A_1 e^{\lambda_{i_1} t} + A_2 e^{\lambda_{i_2} t}$$

$$\lambda_{i_{1,2}} = -\gamma \pm \sqrt{\gamma^2 - \kappa'_i} \tag{6}$$

*where $A_1$ and $A_2$ are constants.*

By examining the analogous gradient flow for the vanilla GD, where the training error decays at the rate $e^{-\kappa_i t}$ (Wang et al., 2021), thus PINNs under vanilla GD suffer from *spectral bias*.

Once we add momentum, the decay rate analysis becomes more involved as Eq. 6 yields three different cases of solutions. Each of the three cases is analogous to one of the solutions of a damped harmonic oscillator Qian (1999); Arya (Fig. 1):

- Under-damped: Imaginary roots ($\gamma^2 < \kappa'_i$)
- Critically-damped: Real and equal roots ($\gamma^2 = \kappa'_i$)

- Over-damped: Real roots ($\gamma^2 > \kappa_i'$).

**Under-damped case** As $\sqrt{\gamma^2 - \kappa_i'}$ has imaginary roots, Eq. 6 can be rewritten as:

$$Ae^{-\gamma t} \cos\left(\omega_1 t + \phi\right)$$

where $\omega_1 = \sqrt{\kappa_i' - \gamma^2}$, and $A$ and $\phi$ are two constants corresponding to the amplitude and the phase of the damped oscillation. In physics, this solution corresponds to an oscillatory motion in which the amplitude is decaying exponentially.

**Critically-damped case** The general solution of the critically-damped case can be written as:

$$(B_1 + B_2 t)e^{-\gamma t}$$

where $B_1$ and $B_2$ are constants. As the oscillation motion is not present, the decaying rate for this case is much faster (Fig. 1).

**Over-damped case** Lastly, in the over-damped case the general solution is simplified as:

$$e^{-\gamma t}\left(C_1 e^{\omega_2 t} + C_2 e^{-\omega_2 t}\right)$$

where $\omega_2 = \sqrt{\gamma^2 - \kappa_i'}$ and $C_1$ and $C_2$ are constants. Similar to the critically-damped case, the above equation states a fast decay.

Thus, depending on $|\kappa_i|$, the absolute value of the eigenvalues of the kernel matrix $\boldsymbol{K}$, the dynamics of the training error corresponding to different frequency components can differ. For larger eigenvalues, the training error corresponds to an under-damped solution, in which the amplitude of an oscillatory motion is decaying exponentially, whereas for smaller eigenvalues the correspondence is with an over-damped or critically-damped oscillation, with a much faster decay. Thus, when using GDM instead of vanilla GD, the training process for the components of the target function that correspond to the smaller eigenvalues will decay fast (undergoing the over-damped or critically-damped motions), while the components that correspond to the larger eigenvalues will decay at a slower rate. As a consequence, the effect of spectral bias will be less prominent (compared to vanilla GD). To provide visualizations, in Appendix F the decay dynamics of GDM and vanilla GD are plotted using a small and large eigenvalue.

## 4 NUMERICAL EXPERIMENTS

In the previous Section, we proved that for infinitely-wide networks GDM and Adam can diminish the effect of spectral bias in theory. Here, we show that in practice, for sufficiently wide networks, Adam and GDM can significantly diminish the effect of spectral bias. Using Poisson's equation, the transport function, and the reaction-diffusion problem we provide results from our numerical experiments. Experiments for the reaction-diffusion problem are presented in Appendix H.2.

### 4.1 POISSON'S EQUATION

Poisson's equation is a well-known elliptic PDE in physics. For example, in electromagnetism, the solution to Poisson's equation is the potential field of a given electric charge. In Eq. 4 the general form of the one-dimensional Poisson's equation was presented. Here, we write Poisson's equation for a specific source function and a specific boundary condition:

$$f(x) = -C^2 \sin(Cx), \ \ x \in [0,1]$$
$$g(x) = 0, \ \ x = 0, 1$$

The defined loss function as well as the analytical solution of Poisson's equation are shown in Appendix G. We trained a network of two hidden layers of width 500, $N_r = 100$ and $N_b = 100$. Of note, Wang et al. (2022) had shown that during the training process, the NTK of networks with a width of 500 practically stayed constant.

Our numerical experiments confirmed that for a relatively small value of $C = 5\pi$ (where the effect of the spectral bias is not significant), after 55000 epochs, models trained via all three algorithms could accurately estimate the solution (Fig. 2; top panel). The relative error $\|(u - \hat{u})/\hat{u}\|$ for vanilla

GD was on the order of $10^{-2}$. The relative error using GDM and Adam were respectively on the order of $10^{-3}$ and $10^{-4}$ (Fig. 3a). As the parameters of a PDE increase, its solution contains higher frequency modes, and as a result, convergence becomes more challenging. When $C = 10\pi$, after 55000 epochs, the solution obtained from training the network with vanilla GD was far from the exact solution (Fig. 2; bottom panel), and it exhibited a relative error larger than $10^{-1}$ (Fig. 3b). Meanwhile, the trained models with GDM and Adam had relative errors of $10^{-2}$ and $10^{-4}$, respectively (Fig. 3b).

It is also of interest to investigate the behavior of the training loss function and its decay rate. For the low-frequency case ($C = 5\pi$), the training loss via Adam had a faster convergence, however, all three networks converged after about 30000 epochs (Fig. 4, left panel). For the high-frequency case ($C = 10\pi$), the training loss via GD had a much slower decay rate, and after 35000 epochs it did not converge. On the other hand, the training loss for both GDM and Adam could converge, and the model trained via Adam converged after about 25000 epochs (Fig. 4, right panel).

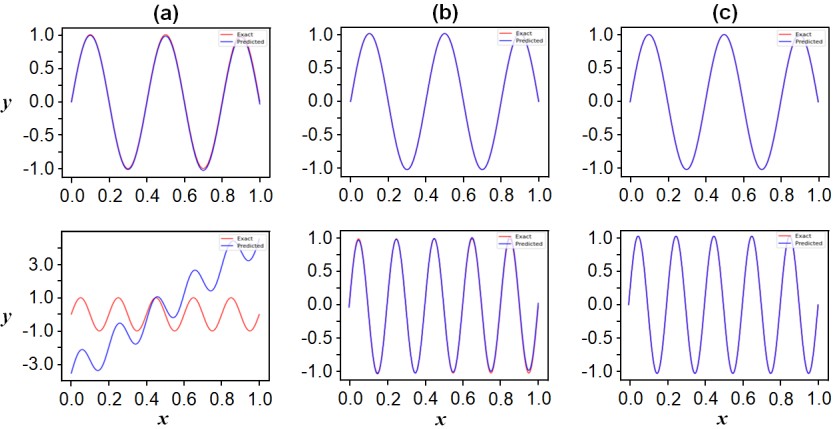

Figure 2: 1D Poisson equation when $C = 5\pi$ (top panel), and $10\pi$ (bottom panel). (a) Vanilla GD after (b) GDM (c) Adam.

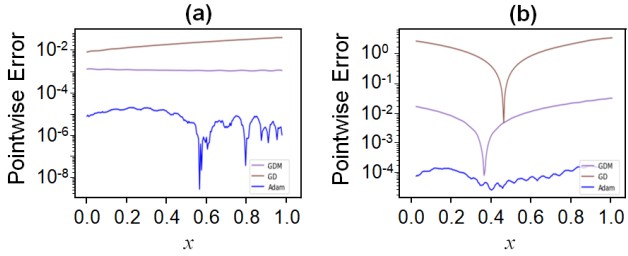

Figure 3: Estimated training error for 1D Poisson equation. a) $C = 5\pi$. b) $C = 10\pi$.

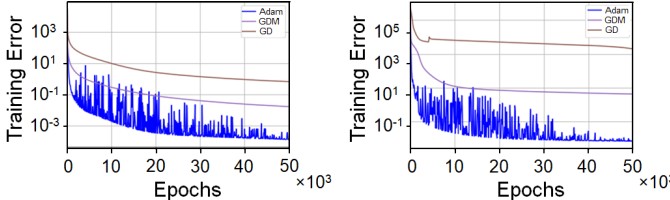

Figure 4: Left panel: training loss when $C = 5\pi$. Right panel: training loss when $C = 10\pi$.

Of note, using vanilla GD and training the model for 180000 epochs resulted in a small relative error on the order of $10^{-2}$ (Fig. G.1). This confirmed our discussion presented in Section 3.1 that vanilla

GD will converge under large parameters, though due to the presence of high-frequency features it is extremely slow.

## 4.2 EVOLUTION OF THE SOLUTION DURING LEARNING

Knowing the fundamental difference between GDM and GD, it is of interest to investigate how under these two algorithms solutions are evolved (in time). Thus, for the Poisson equation when $C = 15\pi$, we have plotted the solutions at epochs 2000, 10000, 20000, 30000, 40000, and 50000 (GD: Fig. 5 and GDM: Fig. 6). For GD: at epoch 10000, the solution has the correct sinusoidal shape, however it is vertically shifted and clearly cannot satisfy the boundary conditions. Plotting the eigenvalues of $\boldsymbol{K}_{bb}$ (Fig. F.1) confirms that they are much smaller than the eigenvalues of $\boldsymbol{K}_{rr}$ indicating that under GD the boundary and initial conditions are learnt much slower. As the training continues in time, the solution becomes less vertically shifted and closer to the boundary condition values. However, the learning process is slow, and even at epoch 50000 the estimation is far from the particular solution. Meanwhile, GDM is much faster, and at epoch 10000 the solution already has the correct sinusoidal shape and satisfies one end of the initial conditions. By epoch 20000 the estimated solution has a low error on the order of $10^{-2}$.

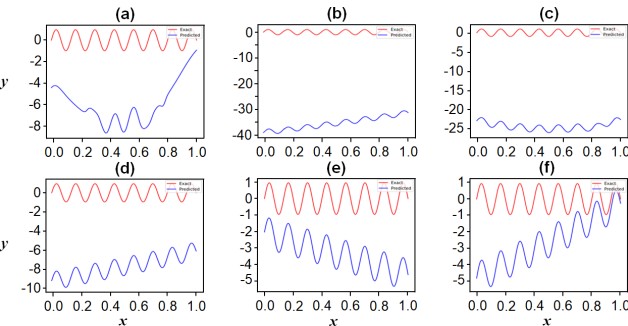

Figure 5: The solution in different stages of the learning process via GD. a) 2000 epochs, b) 10000 epochs, c) 20000 epochs, d) 30000 epochs, e) 40000 epochs, and f) 50000 epochs.

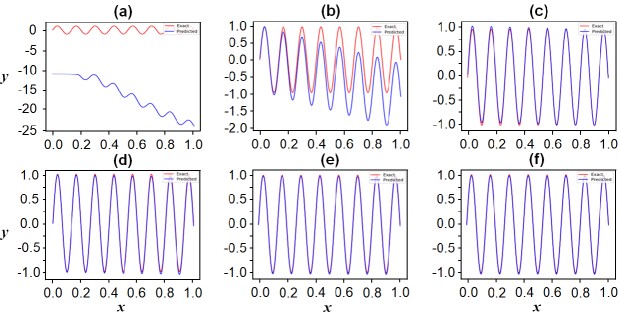

Figure 6: The solution in different stages of the learning process via GDM. a) 2000 epochs, b) 10000 epochs, c) 20000 epochs, d) 30000 epochs, e) 40000 epochs, and f) 50000 epochs.

## 4.3 TRANSPORT EQUATION

The transport equation is a hyperbolic PDE that models the concentration of a substance flowing in a fluid. Here, we focus on a one-dimensional transport equation:

$$\frac{\partial u}{\partial t} + \beta \frac{\partial u}{\partial x} = 0, \;\; x \in \Omega, T \in [0,1]$$
$$g(x) = u(x,0), \;\; x \in \Omega$$

where $\beta$ is a control parameter (independent of $x$ and $t$). To facilitate later comparisons with Krishnapriyan et al. (2021), we used the same network architectures as their study: $N_b = 100$, $N_r = 1500$, and a 4-layer network. We also chose the boundary and initial conditions to be $u(x, 0) = \sin(x)$ and $u(0, t) = u(2\pi, t)$. The defined loss function as well as the analytical solution of the transport function are shown in Appendix G.

Similar to Poisson's equation, for small $\beta$ values, the models trained via vanilla GD, GDM, and Adam could all easily converge to the solution, and had small relative errors. However, for $\beta = 20$, after 125000 epochs the model trained with vanilla GD still failed to find the solution, and the averaged relative error stood at a large value (on the order of $10^0$). However, the model trained via GDM after 55000 epochs could converge to the solution and the estimated solution had the relative error on the order of $10^{-2}$. The estimated solution from training the model via Adam, after only 15000 epochs, had a small relative error also on the order of $10^{-2}$. The exact solution, the estimated solutions (based on the three opti5izers), and the absolute difference between the exact and estimated solutions are shown in Fig. 7.

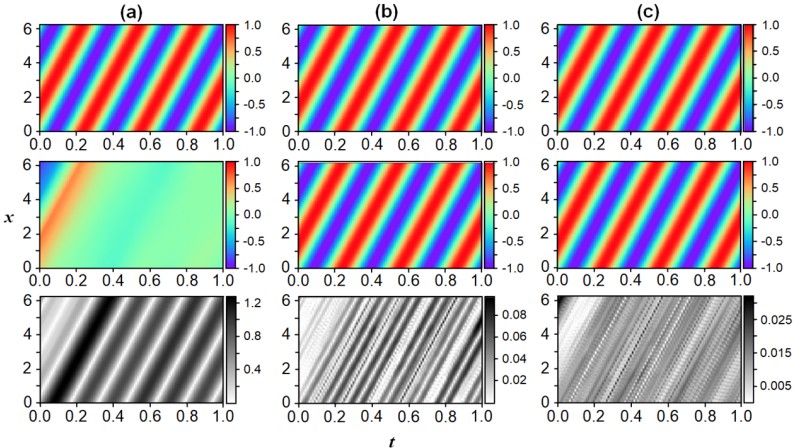

Figure 7: 1D transport equation when $\beta = 20$: Top panels: The exact (analytical) solution. Middle panels: The estimated solutions. Bottom panels: The absolute difference between the exact and estimated solutions. (a) Vanilla GD, (b) GDM, (c) Adam.

## 4.4 COMPARISON WITH OTHER METHODS

Here, we compare the solutions for Poisson's equation and the transport function (from a wide network trained with Adam) with the solutions based on the curriculum learning (hereafter C-learning) Krishnapriyan et al. (2021) and the adaptive weight approach (hereafter AW) presented in Wang et al. (2022). Of note, C-learning used the Limited-memory Broyden–Fletcher–Goldfarb–Shanno (L-BFGS) algorithm that mimics the second-order optimization, as Krishnapriyan et al. (2021) reported that, for their approach, using other optimization algorithms under performed compared to L-BFGS. To reproduce the results we used the code respectively presented in Krishnapriyan et al. (2021) and Wang et al. (2022). All networks were trained with the same architecture introduced in Section (4).

We tested a 1D Poisson's equation with different values of $C$. The comparison between the relative errors of the estimated solutions is plotted in Fig. 8a. For $C \geq 8\pi$, both AW and C-learning exhibited errors of the order of $10^{-1}$. The errors using Adam were at most $10^{-3}$. For the 1D transport function with initial condition $\sin^2(x)$ and for different values of $\beta \in [2, 20]$, the relative error of the estimated solutions using the mentioned methods are plotted in Fig. 8b. As the values of $\beta$ become larger, the estimated solutions from C-learning and AW became less accurate. C-learning for $\beta \geq 15$ and AW for $\beta \geq 10$ had relative errors of the estimated solutions larger than $10^{-1}$. Furthermore, using the C-learning methodology, in order to solve the transport function for $\beta \geq 5$, we had to train models for $\beta = 9, 11, 13, 15, 17, 18, 19, 20$ such that the optimized weights for the previous $\beta$

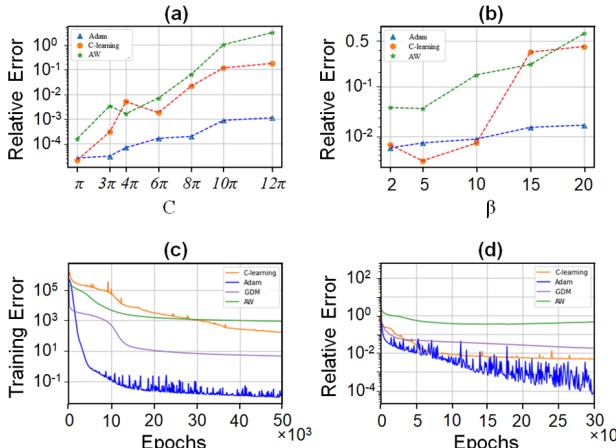

Figure 8: Top Panel: The relative errors of the estimated solutions of 1D transport and Poisson's equations for different values of $\beta$ and $C$ using the C-learning, AW, and Adam. Bottom Panel: the training loss of different methods.

values were used to warm initialize the model for the next beta value. Of note, further comparisons with other choices of initial condition are presented in Appendix H.3.

To further evaluate the effect that the optimization algorithms have on the spectral bias, we compared the rate of training loss decay for wide GDM, wide Adam, AW, and C-learning. An experiment with Poisson's equation with $C = 10\pi$ showed the training loss of the networks based on Adam had much faster decay than AW (see Fig. 8d).

For the transport function with $\beta = 20$, the training loss under Adam converged much faster than the loss under AW, C-learning, and GDM Fig. 8c. Another experiment with Poisson's equation when $C = 10\pi$ showed the training loss of the network based on Adam had a faster decay than AW (see Fig. 8d). Although C-learning also showed a fast training loss convergence, the relative error of the estimate from C-learning was large (see Fig. 8b). This is not surprising, as the loss landscape for PINNsof PDEs with high-frequency modes is compound Krishnapriyan et al. (2021), and they contain many saddle points which attract second-order optimization algorithms Dauphin et al. (2014). Our observation is consistent with (Markidis, 2021), where they also observed this downside of L-BFGS and recommended using Adam for at least the first few epochs. It is also important to mention that, unlike the other methods, the weight initialization for C-learning was not random. Instead, it was based on the optimal solution achieved for lower frequency modes, which requires extra training.

## 5 CONCLUSION

In the present study, through the lens of NTKs, we examined the dynamics of training PINNs via GDM. We also showed that under GDM the convergence rate of low-frequency features becomes slower (analogous to under-damped oscillation) and many high-frequency features undergo much faster conversion dynamics (analogous to over-damped or critically-damped oscillations). Thus, the effect of spectral bias becomes less prominent. Moreover, we discussed how training a PINN via Adam can even further accelerate convergence, and showed it can be much faster than GDM. Although we analyzed the theoretical dynamics of convergence by assuming an infinitely-wide network, our experiments confirmed the estimated solutions obtained from the trained models via GDM and Adam had high accuracy, using finite and practical widths.

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

## A  CONVERGENCE OF WIDE PINNS

**Proposition A.** *Let $\lambda_{b_{\max}}$ and $\lambda_{r_{\max}}$, respectively, be the largest eigenvalues of the Hessians $\nabla^2 \mathcal{L}_b(\boldsymbol{w}^t)$ and $\nabla^2 \mathcal{L}_r(\boldsymbol{w}^t)$. Consider an infinitely-wide PINN optimized with the following update rule:*

$$\boldsymbol{w}_{t+1} = \boldsymbol{w}_t - \eta \nabla \mathcal{L}(\boldsymbol{w}_t),$$

*where $\eta$ is a constant learning rate. Then, provided $\eta = \mathcal{O}(1/(\lambda_{b_{\max}} + \lambda_{r_{\max}}))$, the $\mu$-PL$_*$ condition is satisfied and the PINN will converge.*

*Proof.* The loss function at time $t + 1$ can be written as:

$$\mathcal{L}(\boldsymbol{w}_{t+1}) \approx \mathcal{L}(\boldsymbol{w}_t) + (\boldsymbol{w}_{t+1} - \boldsymbol{w}_t)^\top \nabla \mathcal{L}(\boldsymbol{w}_t) + \frac{1}{2}(\boldsymbol{w}_{t+1} - \boldsymbol{w}_t)^\top \mathcal{H}(\boldsymbol{w}_t)(\boldsymbol{w}_{t+1} - \boldsymbol{w}_t)$$

where $\mathcal{H}(\boldsymbol{w}_t) = \nabla^2 \mathcal{L}_b(\boldsymbol{w}_t) + \nabla^2 \mathcal{L}_r(\boldsymbol{w}_t)$ (Wang et al., 2021). Following the approach of (Wang et al., 2021), let $\boldsymbol{Q}$ be an orthogonal matrix diagonalizing $\mathcal{H}(\boldsymbol{w}_t)$ and $\boldsymbol{v} = \mathcal{L}(\boldsymbol{w}_t)/\|\mathcal{L}(\boldsymbol{w}_t)\|$. With $\boldsymbol{y} = \boldsymbol{Qv}$, we have the following:

$$\mathcal{L}(\boldsymbol{w}_{t+1}) \approx \mathcal{L}(\boldsymbol{w}_t) - \eta \|\nabla L(\boldsymbol{w}_t)\|^2 + \frac{\eta^2}{2} \|\nabla \mathcal{L}(\boldsymbol{w}_t)\|^2 \left( \sum_i^{N_b} \lambda_{b_i} {y_i}^2 + \sum_i^{N_r} \lambda_{r_i} {y_i}^2 \right)$$

$$\lesssim \mathcal{L}(\boldsymbol{w}_t) - \eta \|\nabla \mathcal{L}(\boldsymbol{w}_t)\|^2 + \frac{\eta^2}{2} \|\nabla \mathcal{L}(\boldsymbol{w}_t)\|^2 (\lambda_{b_{\max}} + \lambda_{r_{\max}})$$

where the $\lambda_{b_i}$ and $\lambda_{r_i}$ are the respective eigenvalues of the Hessians of $\mathcal{L}_b$ and $\mathcal{L}_r$ ordered non-decreasingly, and the summation is taken over the components of $\boldsymbol{y}$.

From here, fixing any bound $\mathcal{B}$ such that $\lambda_{b_{\max}} + \lambda_{r_{\max}} \leq \mathcal{B}$, we obtain the following inequality:

$$\mathcal{L}(\boldsymbol{w}_{t+1}) \lesssim \mathcal{L}(\boldsymbol{w}_t) - \eta \|\nabla \mathcal{L}(\boldsymbol{w}_t)\|^2 \left( 1 - \frac{\eta \mathcal{B}}{2} \right)$$

Witness that if $\eta = 1/\mathcal{B}$, we can further simplify this to:

$$\mathcal{L}(\boldsymbol{w}_{t+1}) \lesssim \mathcal{L}(\boldsymbol{w}_t) - \eta \|\nabla \mathcal{L}(\boldsymbol{w}_t)\|^2$$

Since our NN satisfies the $\mu$-PL$_*$ condition from Eq. 3 due to its width, we therefore have:

$$\mathcal{L}(\boldsymbol{w}_{t+1}) \leq (1 - \eta \mu)\mathcal{L}(\boldsymbol{w}_t),$$

which concludes the proof. $\qquad\square$

## B    Hessian for Poisson's Equation

**Proposition B.** *Let $\mathcal{H}$ be the Hessian of the loss function of an infinitely-wide PINN for the one-dimensional Poisson equation defined by Eq. 4 with $u(x) = \sin(Cx)$. Then, we have $\mathcal{H} = \mathcal{O}(C^4)$.*

*Proof.* The Hessian of loss on the collocation points is calculated following the methods introduced in Wang et al. (2021), where the gradient of the loss function is:

$$\frac{\partial \mathcal{L}_r}{\partial w} = \frac{\partial \int_0^1 (\frac{\partial^2 u_w}{\partial x^2} - \frac{\partial^2 u}{\partial x^2})^2 dx}{\partial w}.$$

Here, $w \in \boldsymbol{w}$, $u(x)$ is the target solution (admitting some parameter $C$), and $u_w(x)$ is the NN approximation of the output. Assuming that the approximation is a good representation of the actual solution, it can be written as $u_w(x) = u(x)\epsilon_w(x)$, where $\epsilon_w(x)$ is a smooth function taking values in $[0, 1]$, such that $|\epsilon_w(x) - 1| < \delta$ for some $\delta > 0$ and $\|\frac{\partial^k \epsilon_w(x)}{\partial x^k}\| \leq \delta$, where we have the $L^\infty$ norm.

The Hessian of the loss function will therefore be:

$$\frac{\partial^2 \mathcal{L}_r}{\partial w^2} = \frac{\partial^2 \int_0^1 (\frac{\partial^2 u_w}{\partial x^2} - \frac{\partial^2 u}{\partial x^2})^2}{\partial w^2} = \frac{\partial I}{\partial w}$$

where $I = \frac{\partial \mathcal{L}_r}{\partial w}$, and can be calculated to be:

$$I = 2\frac{\partial^2 u_w}{\partial w \partial x} \left( \frac{\partial^2 u_w}{\partial x^2} - \frac{\partial^2 u}{\partial x^2} \right) \Big|_0^1 - \frac{\partial u_w}{\partial w} \left( \frac{\partial^3 u_w}{\partial x^3} - \frac{\partial^3 u}{\partial x^3} \right) \Big|_0^1 + \int_0^1 \frac{\partial u_w}{\partial w} \left( \frac{\partial^4 u_w}{\partial x^4} - \frac{\partial^4 u}{\partial x^4} \right) dx.$$

The above equation contains 3 terms, which we call $I_1$, $I_2$, and $I_3$ respectively. Note that $\frac{\partial I}{\partial w} = \frac{\partial I_1}{\partial w} + \frac{\partial I_2}{\partial w} + \frac{\partial I_3}{\partial w}$. We calculate these terms as follows:

$$I_1 = 2\frac{\partial^2 u_w}{\partial w \partial x}\left(\frac{\partial^2 u_w}{\partial x^2} - \frac{\partial^2 u}{\partial x^2}\right)\Big|_0^1$$

$$\frac{\partial I_1}{\partial w} = 2\frac{\partial(\frac{\partial^2 u_w}{\partial w \partial x})}{\partial w}(\frac{\partial^2 u_w}{\partial x} - \frac{\partial^2 u}{\partial x^2})\left(\frac{\partial^2 u_w}{\partial x^2} - \frac{\partial^2 u}{\partial x^2}\right)\Big|_0^1$$

$$= 2\left(\frac{\partial\frac{\partial(u'(x)\epsilon_w(x) + u(x)\epsilon_w'(x))}{\partial w}}{\partial w}\right)\left(\frac{\partial^2 u_w}{\partial x^2} - \frac{\partial^2 u}{\partial x^2}\right)\Big|_0^1$$

$$= 2\left(\frac{\partial(u'(x)\frac{\partial\epsilon_w(x)}{\partial w} + u(x)\frac{\partial\epsilon_w'(x)}{\partial w})}{\partial w}\right)\left(\frac{\partial^2 u_w}{\partial x^2} - \frac{\partial^2 u}{\partial x^2}\right)\Big|_0^1$$

$$= 2\left(u'(x)\frac{\partial^2\epsilon_w(x)}{\partial w^2} + u(x)\frac{\partial^2\epsilon_w'(x)}{\partial w^2}\right)\left(\frac{\partial^2 u_w}{\partial x^2} - \frac{\partial^2 u}{\partial x^2}\right)\Big|_0^1.$$

Note that $|u(x)| \leq 1$ and by the chain rule $|u'(x)| \leq C$. An application of the triangle inequality then yields:

$$\left|u'(x)\frac{\partial^2\epsilon_w(x)}{\partial w^2} + u(x)\frac{\partial^2\epsilon_w'(x)}{\partial w^2}\right| \leq C\left\|\frac{\partial^2\epsilon_w(x)}{\partial w^2}\right\| + \left\|\frac{\partial^2\epsilon_w'(x)}{\partial w^2}\right\|.$$

Using the assumptions on $\epsilon_w(x)$ we get:

$$\left|\frac{\partial^2 u_w}{\partial x^2} - \frac{\partial^2 u}{\partial x^2}\right| \leq (C^2 + 2)\delta.$$

Combing both of these, we see $\frac{\partial I_1}{\partial w} = \mathcal{O}(C^3)$. Similarly, $\frac{\partial I_2}{\partial w} = \mathcal{O}(C^3)$ and $\frac{\partial I_3}{\partial w} = \mathcal{O}(C^4)$, which concludes the proof. □

## C GRADIENT FLOW FOR GD WITH MOMENTUM

**Theorem C.** *For an infinitely-wide PINN, the gradient flow of GDM is:*

$$m\begin{bmatrix}\ddot{u}(\boldsymbol{x}_b, \boldsymbol{w}(\boldsymbol{t})) \\ D[\ddot{u}](\boldsymbol{x}_r, \boldsymbol{w}(\boldsymbol{t}))\end{bmatrix} = -\mu\begin{bmatrix}\dot{u}(\boldsymbol{x}_b, \boldsymbol{w}(\boldsymbol{t})) \\ D[\dot{u}](\boldsymbol{x}_r, \boldsymbol{w}(\boldsymbol{t}))\end{bmatrix} - \boldsymbol{K}\begin{bmatrix}u(\boldsymbol{x}_b, \boldsymbol{w}(\boldsymbol{t})) - g(\boldsymbol{x}_b) \\ D[u](\boldsymbol{x}_r, \boldsymbol{w}(\boldsymbol{t})) - h(\boldsymbol{x}_r)\end{bmatrix}$$

*that is analogous to a point mass $m$ undergoing a damped harmonic oscillation in a viscous medium with a friction coefficient of $\mu(\alpha)$ that is function of $\alpha$, Furthermore, $\boldsymbol{K}$ is defined as:*

$$\boldsymbol{K} = \begin{bmatrix}\boldsymbol{K}_{bb} & \boldsymbol{K}_{rb} \\ \boldsymbol{K}_{br} & \boldsymbol{K}_{rr}\end{bmatrix},$$

*where:*

$$\boldsymbol{K}_{bb_{(x,x')}} = \nabla_w u(\boldsymbol{w}, \boldsymbol{x})^\top \nabla_w u(\boldsymbol{w}, \boldsymbol{x}')$$

$$\boldsymbol{K}_{br_{(x,x')}} = \nabla_w u(\boldsymbol{w}, \boldsymbol{x})^\top \nabla_w D[u](\boldsymbol{w}, \boldsymbol{x}')$$

$$\boldsymbol{K}_{rr_{(x,x')}} = \nabla_w D[u](\boldsymbol{w}, \boldsymbol{x}')^\top \nabla_w D[u](\boldsymbol{w}, \boldsymbol{x}')$$

*are three NTKs associated with the boundary and residual terms. Moreover, let $\gamma = \mu/2m$, $\kappa_i$ be the $i$-th eigenvalue of $\boldsymbol{K}$, and $\kappa_i' = \frac{\kappa_i}{m}$. Then, the solutions to the gradient flow are of the form:*

$$A_1 e^{\lambda_{i_1} t} + A_2 e^{\lambda_{i_2} t}$$
$$\lambda_{i_{1,2}} = -\gamma \pm \sqrt{\gamma^2 - \kappa_i'} \tag{7}$$

*where $A_1$ and $A_2$ are constants.*

*Proof.* Recall that as the NN becomes wider, the norm of the Hessian becomes smaller, such that in the limit as $N \to \infty$ the norm of Hessian becomes 0. One immediate consequence of small Hessian for a NN is that its output can be estimated by a linear function Lee et al. (2019). The output of a NN can thus be replaced by its first-order Taylor expansion:

$$u_t^{\text{lin}}(\boldsymbol{w}) \approx u(\boldsymbol{w})|_{\boldsymbol{w}_0} + (\boldsymbol{w} - \boldsymbol{w}_0)\nabla u(\boldsymbol{w})|_{\boldsymbol{w}_0}.$$

The update rule for GDM can be written as Du (2019):

$$\boldsymbol{w}_{t+1} = \boldsymbol{w}_t + \alpha(\boldsymbol{w}_t - \boldsymbol{w}_{t-1}) - \eta\nabla_w\mathcal{L}(\boldsymbol{w}).$$

The discrete updates to the output of NN become (see Appendix C.1):

$$u_{t+1}^{\text{lin}} = u_t^{\text{lin}} + \alpha(u_t^{\text{lin}} - u_{t-1}^{\text{lin}}) - \eta\nabla_w\mathcal{L}(\boldsymbol{w})\nabla_w u(\boldsymbol{w})|_{\boldsymbol{w}_0}$$

In the rest of this section, for simplicity, we will drop the "lin" term. The dynamics of GDM are analogous to the equation of motion of a point mass $m$ undergoing a damped harmonic oscillation (Appendix C.2):

$$m\ddot{u} + \mu\dot{u} - \nabla_w\mathcal{L}(\boldsymbol{w})f(u) = 0$$

where $f(u)$ is a linear function of $u$, and $\mu$ is the friction coefficient that is related to the momentum term in GDM as such: $\alpha = \frac{m}{m+\mu\Delta t}$ Qian (1999). Thus, the gradient flow of $u_t$ and $D[u](x, w(t))$ can be written as:

$$\begin{aligned}
m\ddot{u}(\boldsymbol{x}_b, \boldsymbol{w(t)}) &= -\mu\dot{u}(\boldsymbol{x}_b, \boldsymbol{w(t)}) - \boldsymbol{K}_{bb_{(x,x')}}(\boldsymbol{w})(u(\boldsymbol{x}_r, \boldsymbol{w(t)}) - g(\boldsymbol{x}_b)) \\
&\quad - \boldsymbol{K}_{rb_{(x,x')}}(\boldsymbol{w})(D[u](\boldsymbol{x}_r, \boldsymbol{w(t)}) - h(\boldsymbol{x}_r)) \\
mD[\ddot{u}](\boldsymbol{x}_r, \boldsymbol{w(t)}) &= -\mu D[\dot{u}](\boldsymbol{x}_r, \boldsymbol{w(t)}) - \boldsymbol{K}_{br_{(x,x')}}(\boldsymbol{w})(u(\boldsymbol{x}_b, \boldsymbol{w(t)}) - g(\boldsymbol{x}_b)) \\
&\quad - \boldsymbol{K}_{rr_{(x,x')}}(\boldsymbol{w})(D[u](\boldsymbol{x}_r, \boldsymbol{w(t)}) - h(\boldsymbol{x}_r)).
\end{aligned} \tag{C.1}$$

As mentioned earlier, in wide NNs if the last layer of the network is linear then the tanget kernels are static.

We write Eq. C.1 in matrix form as follows:

$$m\begin{bmatrix}\ddot{u}(\boldsymbol{x}_b, \boldsymbol{w(t)}) \\ D[\ddot{u}](\boldsymbol{x}_r, \boldsymbol{w(t)})\end{bmatrix} = -\mu\begin{bmatrix}\dot{u}(\boldsymbol{x}_b, \boldsymbol{w(t)}) \\ D[\dot{u}](\boldsymbol{x}_r, \boldsymbol{w(t)})\end{bmatrix} - \boldsymbol{K}\begin{bmatrix}u(\boldsymbol{x}_b, \boldsymbol{w(t)}) - g(\boldsymbol{x}_b) \\ D[u](\boldsymbol{x}_r, \boldsymbol{w(t)}) - h(\boldsymbol{x}_r)\end{bmatrix} \tag{C.2}$$

As $\boldsymbol{K}$ is a positive semi-definite matrix Wang et al. (2021) Eq. C.2 can be viewed as a set of independent differential equations, each one corresponding to an eigenvalue $\lambda_i$ of the kernel. These give rise to the individual general solutions of the form:

$$A_1 e^{\lambda_{i_1}t} + A_2 e^{\lambda_{i_2}t}$$

$$\lambda_{i_{1,2}} = -\gamma \pm \sqrt{\gamma^2 - \kappa_i'}$$

where $A_1$ and $A_2$ are constants, $\gamma = \mu/2$, and $\kappa_i' = \frac{\kappa_i}{m}$. Of note, $\mu$ and $m$ are set by the user, which in turn define the value of momentum. The choice of values of $m$ and $\mu$ will determine the rate of decay of the above equation. $\qquad\square$

## C.1 Linear NN Updates

The Taylor expansions of the outputs at time steps $t + 1$ and $t$ are:

$$u_{t+1}^{\text{lin}}(\boldsymbol{w}) = u(\boldsymbol{w})|_{\boldsymbol{w}_0} + (\boldsymbol{w}_{t+1} - \boldsymbol{w}_0)\nabla u(\boldsymbol{w})|_{\boldsymbol{w}_0}$$

$$u_t^{\text{lin}}(\boldsymbol{w}) = u(\boldsymbol{w})|_{\boldsymbol{w}_0} + (\boldsymbol{w}_t - \boldsymbol{w}_0)\nabla u(\boldsymbol{w})|_{\boldsymbol{w}_0}.$$

Thus, the difference between the outputs in the interval is:

$$\begin{aligned}
u_{t+1}^{\text{lin}}(\boldsymbol{w}) - u_t^{\text{lin}}(\boldsymbol{w}) &= \nabla u(\boldsymbol{w})|_{\boldsymbol{w}_0}(\boldsymbol{w}_{t+1} - \boldsymbol{w}_t) \\
&= \nabla u(\boldsymbol{w})|_{\boldsymbol{w}_0}(\alpha(\boldsymbol{w}_t - \boldsymbol{w}_{t-1}) - \eta\nabla\mathcal{L}(\boldsymbol{w}_t))
\end{aligned}$$

where we used the update rule of GD with momentum within the equation. Similarly, we have:

$$u_t^{\text{lin}}(\boldsymbol{w}) - u_{t-1}^{\text{lin}}(\boldsymbol{w}) = \nabla u(\boldsymbol{w})|_{\boldsymbol{w}_0}(\boldsymbol{w}_t - \boldsymbol{w}_{t-1})$$

and:

$$u_{t+1}^{\text{lin}}(\boldsymbol{w}) - u_t^{\text{lin}}(\boldsymbol{w}) = \nabla u(\boldsymbol{w})|_{\boldsymbol{w}_0}\left(\alpha\left(\frac{u_t^{\text{lin}}(\boldsymbol{w}) - u_{t-1}^{\text{lin}}(\boldsymbol{w})}{\nabla u(\boldsymbol{w})|_{\boldsymbol{w}_0}}\right) - \eta\nabla\mathcal{L}(\boldsymbol{w}_t)\right).$$

Thus:

$$u_{t+1}^{\text{lin}}(\boldsymbol{w}) = u_t^{\text{lin}}(\boldsymbol{w}) - \eta\nabla\mathcal{L}(\boldsymbol{w}_t) \cdot \nabla u(\boldsymbol{w})|_{\boldsymbol{w}_0} - \alpha(u_t^{\text{lin}}(\boldsymbol{w}) - u_{t-1}^{\text{lin}}(\boldsymbol{w})).$$

## C.2 Relation between GDM and Damped Oscillation

The dynamics of a point mass $m$ undergoing a damped harmonic oscillation with a friction coefficient of $\mu$ are given by:

$$m\ddot{\boldsymbol{w}} + \mu\dot{\boldsymbol{w}} = -\nabla_w \mathcal{L}(\boldsymbol{w}) \tag{3}$$

where $\nabla_w \mathcal{L}(\boldsymbol{w})$ is the force field. Qian (1999) showed that Eq. 3 in its discrete format can be written as:

$$m\frac{\boldsymbol{w}_{t+\Delta t} + \boldsymbol{w}_{t-\Delta t} - 2\boldsymbol{w}_t}{\Delta t^2} + \mu\frac{\boldsymbol{w}_{t+\Delta t} - \boldsymbol{w}_t}{\Delta t} = -\nabla_w \mathcal{L}(\boldsymbol{w}).$$

After some algebraic simplifications, the above equation becomes:

$$\boldsymbol{w}_{t+\Delta t} - \boldsymbol{w}_t = \frac{m}{m + \mu\Delta t}(\boldsymbol{w}_t - \boldsymbol{w}_{t-\Delta t}) - \frac{\Delta t^2}{m + \mu\Delta t}\nabla_w \mathcal{L}(\boldsymbol{w}).$$

Clearly, $\frac{m}{m+\mu\Delta t}$ is equivalent to the momentum term in GDM, and $\frac{\Delta t^2}{m+\mu\Delta t}$ can be treated as the learning term. The dynamics of the output of a wide network (Appendix C.1) under GDM are similar.

## D Analysis of Adam for Band-Limited Functions

Adam is a commonly-used optimizer which can be interpreted as GDM adapted for variance (Kingma & Ba, 2014; Balles & Hennig, 2018). Thus, it has the same properties as GDM related to the diminishing of the spectral bias because of the momentum term (see Appendix E for more details). However, because of the adaptive learning rate, it can be expected to be even faster than GDM. Here, we briefly provide more insights into the general behavior of Adam.

Optimizing with Adam, we define:

$$g(\boldsymbol{w}) = \sum_{\boldsymbol{x} \in X} \nabla\ell(\boldsymbol{w}; \boldsymbol{x})/M,$$

where $\ell = \mathcal{L}_{b,r}$, $M$ is the batch size and $X$ is the batch. Furthermore, the pointwise variance of $g(\boldsymbol{w})$ is written as $\sigma = \operatorname{var} g(\boldsymbol{w})$. We will use subscripts to denote entries, so that $\sigma_i$ is $i$-th entry of $\sigma$. With $\nabla\mathcal{L}$ being the true gradient, at each epoch, Adam updates $\boldsymbol{w}_i$ with a magnitude inversely proportional to an estimator of $\sigma_i^2/\nabla\mathcal{L}_i^2$ (Balles & Hennig, 2018).

The goal of the training process is to find a weight $\boldsymbol{w}$ which minimizes the loss $\mathcal{L}$ across all collocation and boundary points from Eq. 2. The role that variance adaptation plays in this speedup is readily seen when the solution to a PDE is (or is well-approximated by) a band-limited function. A band-limited function on a finite set of frequencies $k$ has a form of:

$$u(\boldsymbol{x}) = \sum_{k \in K} \alpha_k \exp(2\pi i k \boldsymbol{x}).$$

When using GD, the solutions eventually satisfy the following bound with high probability (Basri et al., 2019; Arora et al., 2019):

$$\mathcal{L} \lessapprox \sqrt{\frac{2\pi \sum_{k \in K} \alpha_k^2 k^2}{N}}.$$

This indicates that a network optimized with GD will learn better approximations of $u(\boldsymbol{x})$ if lower frequencies $k$ are present (especially when compared to another solution with otherwise identical amplitudes $\alpha_k$). Conversely, if $u(\boldsymbol{x})$ is primarily high-frequency then $\mathcal{L}$ has very weak bounds. Since our NN is infinitely-wide it satisfies the $\mu$-PL$_*$ condition $\|\nabla\mathcal{L}\|^2 \geq \mu\mathcal{L}$ of Eq. 3. This implies similarly weak bounds on $\|\nabla\mathcal{L}\|$ and thus $\nabla\mathcal{L}_i$ (since $\mu$ is constant). Furthermore, convergence requires $\nabla\mathcal{L}_i \approx 0$ for all $i$, so if any $\nabla\mathcal{L}_i \gg 0$ the network has yet to converge. However, this is precisely the situation in which Adam accelerates toward the solution fastest as its updates are largest when $\sigma_i^2/\nabla\mathcal{L}_i^2$ close to zero. That is, in the presence of high-frequency features resulting in poor bounds on the loss if optimizing with GD, Adam would instead benefit from accelerated convergence. Combined with the inclusion of momentum as discussed earlier, this may provide the framework for it outperforming even GDM. Our numerical experiments confirm that for sufficiently wide NNs both GDM and Adam can converge to desirable solutions, but that Adam is even faster than GDM (see Section 4).

## E  RELATION BETWEEN ADAM AND DAMPED OSCILLATION

It has been observed that Adam yields similar convergence rates to GDM (Kingma & Ba, 2014). Here, we demonstrate that Adam also has dynamics equivalent to a damped oscillator, similar to GDM (Appendix C.2). First, let us briefly recall the traditional update rule for Adam:

$$\Delta \boldsymbol{w}_t = \frac{-\eta}{\sqrt{\hat{\nu}_t} + \epsilon} \hat{m}_t$$

where $m_t$ and $\nu_t$ are defined as:

$$\hat{m}_t = \frac{m_t}{1 - \beta_1^{t+1}}, \quad m_t = \beta_1 m_{t-1} + (1 - \beta_1) g_t$$

$$\hat{\nu}_t = \frac{\nu_t}{1 - \beta_2^{t+1}}, \quad \nu_t = \beta_2 \nu_{t-1} + (1 - \beta_2) g_t{}^2$$

$$g(t) = -\nabla_w \mathcal{L}(\boldsymbol{w}).$$

**Proposition E.** *The update rule for Adam can be written as:*

$$\boldsymbol{w}_{t+1} = \boldsymbol{w}_t - P(t)(\boldsymbol{w}_t - \boldsymbol{w}_{t-1}) - Q(t) \nabla_w \mathcal{L}(\boldsymbol{w}_t).$$

*where:*

$$P(t) = \frac{\beta_1 (\sqrt{\nu_{t-1}} + \epsilon)(1 - \beta_1(t-1))}{(\sqrt{\nu_t} + \epsilon)(1 - \beta_1 t)}$$

$$Q(t) = \frac{\eta(1 - \beta_1 t)}{(\sqrt{\nu_t} + \epsilon)(1 - \beta_1 t)}.$$

*Proof.* It is straightforward to show that the update rule can be written as:

$$\boldsymbol{w}_{t+1} = \boldsymbol{w}_t - \frac{\eta \beta_1 m_{t-1}}{(\sqrt{\nu_t} + \epsilon)(1 - \beta_1 t)} - \frac{\eta(1 - \beta_1)}{(\sqrt{\nu_t} + \epsilon)(1 - \beta_1 t)} \nabla_w \mathcal{L}(\boldsymbol{w}_t).$$

Inserting $\boldsymbol{w}_t = \boldsymbol{w}_{t-1} - \frac{\eta}{\sqrt{\hat{\nu}_t} + \epsilon} \hat{m}_t$ into the above equation shows:

$$\boldsymbol{w}_{t+1} = \boldsymbol{w}_t - \frac{\beta_1(\sqrt{\nu_{t-1}} + \epsilon)(1 - \beta_1(t-1))}{(\sqrt{\nu_t} + \epsilon)(1 - \beta_1 t)}(\boldsymbol{w}_t - \boldsymbol{w}_{t-1}) - \frac{\eta(1 - \beta_1 t)}{(\sqrt{\nu_t} + \epsilon)(1 - \beta_1 t)} \nabla_w \mathcal{L}(\boldsymbol{w}_t). \quad \text{(E.1)}$$

By the definition of $P(t)$ and $Q(t)$ we can rewrite Eq. E.1 as:

$$\boldsymbol{w}_{t+1} = \boldsymbol{w}_t - P(t)(\boldsymbol{w}_t - \boldsymbol{w}_{t-1}) - Q(t) \nabla_w \mathcal{L}(\boldsymbol{w}_t) \qquad \text{(E.2)}$$

as claimed. □

Clearly, Eq. E.2 has the same format of the GDM update rule. Thus, the weight updates for Adam follows the same dynamics as SGDM, that of oscillatory motion of a point mass under friction (from Appendix C.2).

However, based on the updated value of the second momentum of the gradient of loss function, both the momentum and the learning rate can be updated by Adam at each iteration. Thus, the convergence to the solution by Adam is much faster compared to GDM.

## F  DECAY COMPARISON BETWEEN GDM AND GD

### F.1  NUMERICAL EVALUATION OF DECAY RATE FOR GD AND GDM

Here, the dynamics of the error decay of GDM and vanilla GD under a large eigenvalue ($\lambda_1 = 10^3$) and a relatively small eigenvalue ($\lambda_2 = 10^{-5}$) are plotted. For vanilla GD, the training error decays at a rate of $e^{-\lambda_i t}$. As shown in Fig. E.1a, for the $\lambda_2$ the training error decays slowly, however is very fast for $\lambda_1$ (almost immediately dropping to zero). Thus, components of the target function

that correspond to the large eigenvalues will be learned much faster. These are the eigenvalues corresponding to lower frequencies in the target function, so clearly the network suffers from spectral bias under GD.

In contrast, learning under GDM, the training error for $\lambda_1$ decays following under-damped oscillation (the red curve in Fig. E.1b), while the training error for $\lambda_2$ follows over-damped oscillation (the blue curve in Fig. E.1a). The training error of both eigenvalues, the large $\lambda_1$ and small $\lambda_2$, decay at approximately the same time. The components of the target function that correspond to the larger eigenvalue are thus not learned any faster, so the effect of spectral bias is minimized.

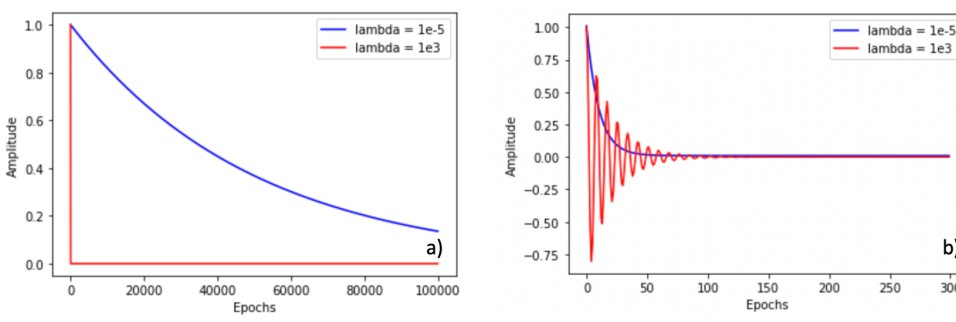

Figure E.1: The comparison of the error decay for a small and a large eigenvalue, a) learning decay during GD optimization process, b) learning decay during GDD optimization process.

### F.2 THE EIGENVALUES OF $\boldsymbol{K}_{bb}$ AND $\boldsymbol{K}_{rr}$

As discussed in Section 3.2 under GD optimization, the total training error is evaluated based on $e^{-\kappa_i t}$, where $\kappa_i$ are the eigenvalues of $\boldsymbol{K}$ that encapsulated $\boldsymbol{K}_{bb}$ (NTK for boundary and initial data points) and $\boldsymbol{K}_{rr}$ (NTK for collocation points representing the PDE). Thus, the convergence rate of the training error is evaluated based on the eigenvalues of $\boldsymbol{K}_{bb}$, and $\boldsymbol{K}_{rr}$ together. Consequently, there might be a discrepancy between the absolute value of eigenvalues of $\boldsymbol{K}_{bb}$, and $\boldsymbol{K}_{rr}$, meaning that eigenvalues of one the two matrices be much larger, and the convergence rate of the training error for them becomes much faster. Hence, the network becomes biased to learn the components corresponding to those eigenvalues first.

In Fig. F.1, for Poisson's equation when $C = 15\pi$, the eigenvalues at initialization for $\boldsymbol{K}_{bb}$(left panel) and $\boldsymbol{K}_{rr}$ (right panel) (in descending order) are plotted. Clearly, the eigenvalues of $\boldsymbol{K}_{rr}$ that represents the PDE, are much larger than the eigenvalues of $\boldsymbol{K}_{bb}$ that represents the initial and boundary data points (bcs). Hence, not surprisingly, in GD where the training errors decay based on $e^{-\kappa_i t}$ the network learns the PDE general form first and becomes slow in learning the bcs. This bias in learning the PDE first can be minimized while we are implementing GDM. In fact, plotting the terms of the loss function (PDE and bcs terms) also confirms that under GD the bcs are learnt slowly (see Fig. F.2) and the rate of decay of loss for bcs (green curve) is much slower than the rate of decay of loss in GDM (blue curve). Thus, clearly, we can see that under GD optimization, the

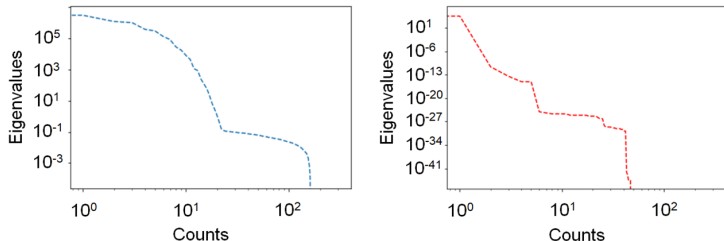

Figure F.1: Left panel: Eigenvalues for $\boldsymbol{K}_{bb}$. Right panel: Eigenvalues for $\boldsymbol{K}_{rr}$

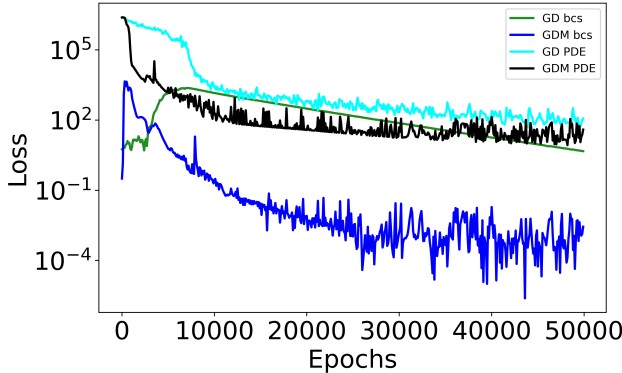

Figure F.2: The loss function during training via GDM and GD for the PDE term of the loss as well as the boundary/initial term are plotted.

network is much slower in learning the bcs (compared to the PDE term). As explained earlier, this is not surprising as the eigenvalues of $\boldsymbol{K}_{bb}$ are much smaller than the eigenvalues of $\boldsymbol{K}_{rr}$.

These observations are very insightful as they reveal a significant difference between PINNs and regular neural networks. Perhaps to explain the difference, it is useful to investigate the evolution of the solution via GD for a completely data-driven case. We trained a fully-connected forward neural network with the same architecture as the above PINN. The training data-set contained 3000 synthetic data points generated based on the solution of Poisson's equation, and we used the mean squared error loss function. The solutions after 5000 and 50000 epochs are shown in Fig. F.3. As the fully-connected network has no physics-informed regularization term, it has difficulties estimating the correct solution. Despite this, it still exhibits spectral bias, as the estimated solutions at both 5000 and 50000 epochs represent low-frequency sinusoidal forms. However, unlike a PINN there is no vertical shift in the solutions. Clearly, the physics-informed regularization term helps significantly to resolve the classical spectral bias (dealing with high-order frequency target functions), learning the correct sinusoidal shape faster. In return, the solutions are worse at satisfying the boundary condition values. This is because the eigenvalues of the boundary kernel represent high-frequency modes.

## G  LOSS FUNCTION OF EQUATIONS

### G.1  LOSS FUNCTION FOR POISSON'S EQUATION

For Poisson's equation:

$$f(x) = -C^2 \sin(Cx), \quad x \in [0,1]$$
$$g(x) = 0, \quad x = 0, 1$$

$u(x) = \sin(Cx)$ is used as the exact solution. The corresponding loss function is written as:

$$\mathcal{L}(\boldsymbol{w}) := \frac{1}{N_b} \sum_{i=1}^{N_b} \left( \hat{u}(x_b{}^i, \boldsymbol{w}) - g(x_b{}^i) \right)^2 + \frac{1}{N_r} \sum_{i=1}^{N_r} \left( \frac{\partial^2 \hat{u}}{\partial x^2}(x_r{}^i, \boldsymbol{w}) - f(x_r{}^i) \right)^2$$

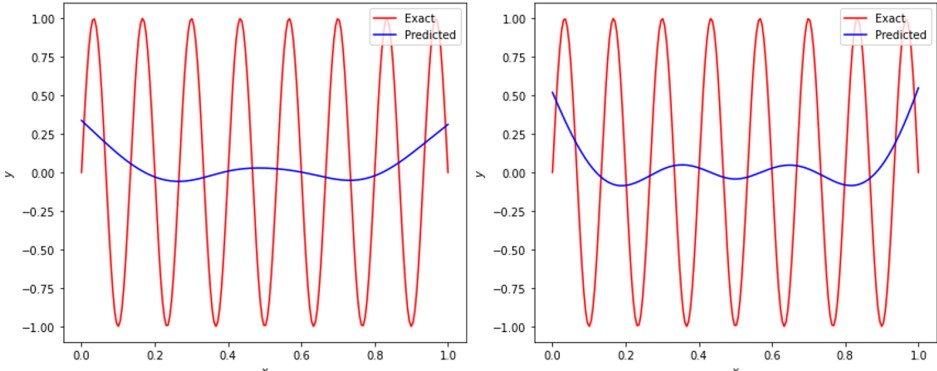

Figure F.3: The completely data-driven solution of the Poisson equation based on GD optimization process. a) after 5000 epochs, b) after 50000 epochs.

where $\hat{u}$ is the output of the network.

### G.2 LOSS FUNCTION FOR THE TRANSPORT FUNCTION

Using the methods of characteristics (Evans, 2010), the transport function has a well-defined analytical solution: $u(x,t) = g(x - \beta t)$. This is used as the exact solution. The corresponding loss function is written as:

$$\mathcal{L}(\boldsymbol{w}) := \frac{1}{N_b} \sum_{i=1}^{N_b} \left( \hat{u}(x_b{}^i, t_b{}^i, \boldsymbol{w}) - g(x_b{}^i) \right)^2 + \frac{1}{N_r} \sum_{i=1}^{N_r} \left( \frac{\partial \hat{u}(x_r{}^i, t_b{}^i, \boldsymbol{w})}{\partial t} + \beta \frac{\partial \hat{u}(x_r{}^i, t_b{}^i, \boldsymbol{w})}{\partial x} \right)^2$$

where $\hat{u}$ is the output of the network.

## H EXPERIMENT PLOTS

### H.1 POISSON EQUATION

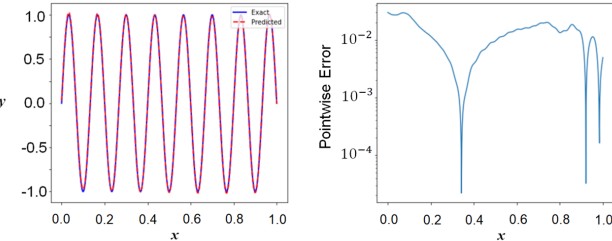

Figure G.1: The estimation of the solution of the Poisson's equation for $C = 15\pi$, using the vanilla GD algorithm after 180000 epochs.

## H.2 REACTION-DIFFUSION EQUATION

A reaction-diffusion equation contains a reaction and a diffusion term. Its general form is written as:

$$\frac{\partial u}{\partial t} = \nu \Delta u + f(u) \tag{G.1}$$

where $u(\boldsymbol{x}, \boldsymbol{t})$ is the solution describing the density/concentration of a substance, $\Delta$ is the Laplace operator, $\nu$ is a diffusion coefficient, and $f(u)$ is a smooth function describing processes that change the present state of $u$ (for example, birth, death or a chemical reaction). Here, we assume a one-dimensional equation, where $f(u) = \rho u(1 - u)$. For $\rho$ independent of $x$ and $t$, and with the defined $f(u)$, Eq. G.1, can be solved analytically (Evans, 2010). To estimate the solution using PINNs, the loss function for the 1D reaction-diffusion PDE is written as:

$$\mathcal{L}(\boldsymbol{w}) := \frac{1}{N_b} \sum_{i=1}^{N_b} \left( \hat{u}(x_b^i, t_b^i, \boldsymbol{w}) - g(x_b^i) \right)^2$$

$$+ \frac{1}{N_r} \sum_{i=1}^{N_r} \left( \frac{\partial \hat{u}}{\partial t}(x_r^i, t_b^i, \boldsymbol{w}) - \nu \frac{\partial^2 \hat{u}}{\partial x^2}(x_r^i, t_b^i, \boldsymbol{w}) - \rho \hat{u}(x_r^i, t_b^i)(1 - \hat{u}(x_r^i, t_b^i)) \right)^2.$$

For consistency with Krishnapriyan et al. (2021), we used $N_r = 1000$, and $N_b = 100$. We also chose the initial and boundary conditions $u(x, 0) = \exp{-\frac{(x-\pi)^2}{\pi^2/2}}$ and $u(0, t) = u(2\pi, t)$ respectively. Similar to the previous experiments, for larger choices of $\nu$ and $\rho$ the model trained via vanilla GD (after 85000 epochs) had difficulty converging to the solution (Fig. G.3, left panel). However, models trained via GDM and Adam (after 45000 epochs) could provide solutions with low error. The plots of estimated and exact solutions for the three algorithms when $\nu = 3$ and $\rho = 5$ are shown in Fig. G.2.

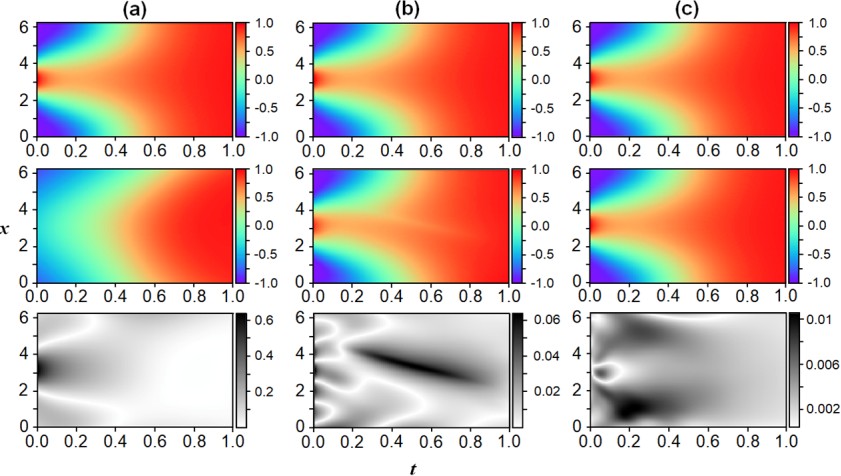

Figure G.2: 1D reaction-diffusion equation for $\nu = 3$ and $\rho = 5$. The solutions are obtained by training a 4-layer network with width=500 at each layer. Top panels: The exact (analytical) solution. Middle panels: The estimated solution. Bottom panels: The absolute difference between the exact and estimated solutions. (a) Vanilla GD, (b) GDM, (c) Adam.

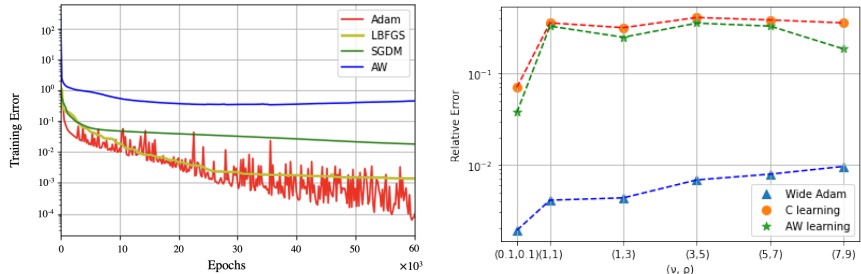

Figure G.3: Left Panel: The training losses of the network, when $\nu = 9$ and $\rho = 7$), via AW, C-learning, wide GDM, and wide Adam are plotted. Right Panel: The relative error of the estimated solutions for different values of $\nu$ and $\rho$ are plotted.

We further compared the solutions of the wide Adam, C-learning, and AW. Similar to the previous experiments, for higher-frequency modes Adam showed superior results. The relative errors of the estimated solutions for different values of $\nu$ and $\rho$ were computed (Fig. 8 (left panel)). Furthermore, when $\nu = 9$ and $\rho = 7$), we observed that the loss function under GDM and Adam had much faster decay than under AW. But, the loss under L-BFGS decayed as fast as under Adam. We reasoned as to why in Section 4.4.

### H.3 Transport function: Further Comparison with different Initial Conditions

We implemented different initial conditions for the transport function and ran several experiments. For the initial condition of $\tanh(x)$, when $\beta \geq 6$ both C-learning and AW estimated solutions of showed large relative errors on the order of $10^{-1}$. However, the estimated solution of a wide network trained via Adam had an acceptable relative error on the order of $10^{-2}$ (Fig. G.4). The estimated solutions for all three optimizers when $\beta = 6$ are shown in Fig. G.5. All networks were trained using a 4-layer network, and AW and Adam both had a width of 500 neurons. The estimated solutions are based on training the model for 85000 epochs.

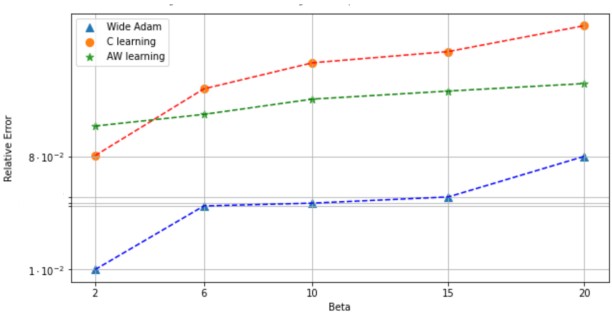

Figure G.4: The relative errors of the estimated solutions of 1D transport function, for different values of $\beta$ using the curriculum learning method (red dashed line), and our approach (blue dashed line) are plotted.

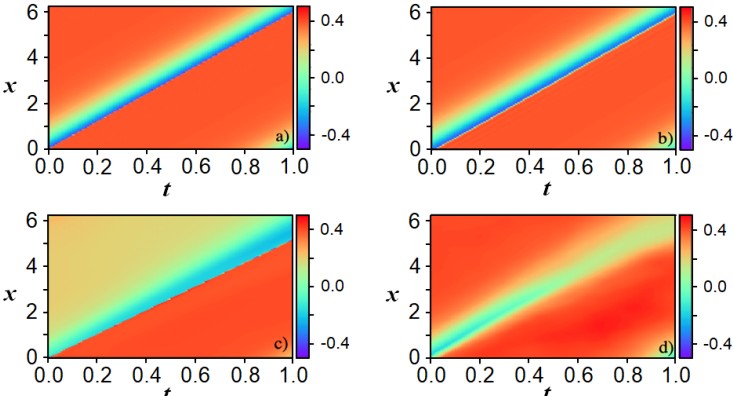

Figure G.5: The estimated solutions of the transport function based on the initial condition of $\tanh(x)$ when $\beta = 6$: a) The exact solution b) Estimated solution of a wide network trained via Adam c) Estimated solution of C-learning d) Estimated solution of AW

For the initial condition of $\sin(x)\cos(x)$, C-learning had small estimated errors (on the order of $10^{-2}$) for $\beta \leq 20$, however for higher values of $\beta$ the error grew larger (Fig. G.6). AW had errors on the order of $10^{-2}$ for $\beta \leq 10$, however the error began to grow for larger values of $\beta$. Adam had errors on the order of $10^{-2}$ even up to $\beta \leq 25$ (Fig. G.6). The estimated solutions based on C-learning, AW, and wide Adam for $\beta = 25$ are shown in Fig. G.7. All networks were trained using a 4-layer network. The estimated solutions are based on training the model for 85000 epochs.

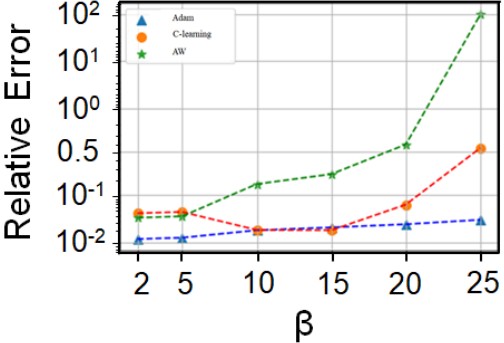

Figure G.6: The relative errors of the estimated solutions of 1D transport function, based on the initial condition of $\sin(x)\cos(x)$, for different values of $\beta$ using the curriculum learning method (red dashed line), and our approach (blue dashed line) are plotted.

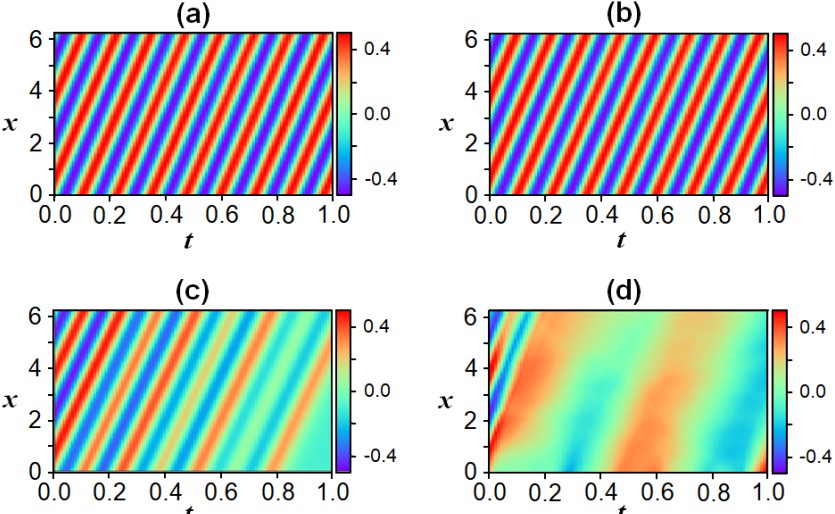

Figure G.7: The estimation of the solution of the transport function based on the initial condition of $\sin(x)\cos(x)$ when $\beta = 25$: a) The exact solution b) Estimated solution of a wide network trained via Adam c) Estimated solution of C-learning d) Estimated solution of AW

