# OpenReview forum: "Momentum Diminishes the Effect of Spectral Bias in Physics-Informed Neural Networks"
_ICLR.cc/2023/Conference — Submitted to ICLR 2023_

### Official Review · Reviewer_3LnV · 2022-10-18

**Confidence:** 3
**Correctness:** 2
**Technical Novelty And Significance:** 3
**Empirical Novelty And Significance:** 3
**Recommendation:** 6

**Clarity, Quality, Novelty And Reproducibility:**

### Quality, Novelty, and Reproducibility:
The paper treats an interesting topic and the perspective about the effects of momentum is definitely novel (to the best of my knowledge). However, some important questions regarding the experimental evidence and possible alternative hypotheses need to be taken care of to increase the paper's quality.

### Clarity:
Generally, the paper is clearly written. Below are pointers to a few exceptions, that can be taken care of easily.
- End of Sec. 2.1: What does "optimal" in "optimal solution $u(x,w)$ refer to? The solution of the PDE, or the minimum of the optimization loss landscape? These may be different.
- Equations at the beginning of Sec. 2.2: There seems to be something wrong. If we insert $u_{h-1}$ into $g_h$, then the weight matrix $\Theta_{h-1}$ is applied twice. Please check.
- In eq. (4), what are $s(x)$ and $b(x)$?
- In Th. 1, what is $m$? Can $m$ be chosen arbitrarily, or does it depend on the parameters of the PDE/the PINN?

### Minor Comments:
- "they are limited to some week empirical evidence" -> weak
- Proposition A -> Proposition 1
- After Th. 1, the references to eq. (7) should be a reference to eq. (6), right?
- In Sec. 4.3, "unperformed" -> underperformed; Section (4) -> Section 4; the results for the Poisson equation with $C=10\pi$ are described twice.

*EDIT* Increased score during discussion period.

**Strength And Weaknesses:**

The paper is well-written and covers an interesting and timely topic. While the authors do not propose a novel method, their experimental evidence is in favor of their hypothesis. The supplementary material contains additional information that helps shedding light on the interplay between the spectral properties of the solution, the selection of the optimizer, and training success.

What prevents me from giving a better recommendation is that some of the experimental evidence is unconvincing, and that -- I believe -- alternative hypotheses first need to be ruled out. Indeed, judging from the results, it appears as if spectral bias is not the problem underlying convergence problems. Looking at, e.g., Fig. E.2 or Fig. 2, we see that SGD manages to learn the high-frequency components, but with a shift, trend, or offset added to it. Therefore, the spectral bias, which suggests that the high-frequency components are learned later, seems not to be a valid explanation. As an alternative hypothesis, it may be that learned solutions $u$ satisfy the PDE. While it is known that trivial (zero) solutions are attractive for PINNs (arXiv:2109.09338,arXiv:2203.13648), I assume that the same holds for non-trivial solutions that satisfy the PDE. Essentially, I argue that the solution $u$ provided by the PINN satisfies the PDE, but not the initial/boundary conditions. This needs to be confirmed or rejected by investigating the loss term $\mathcal{L}_r(w)$ throughout training, and not only the compound loss $\mathcal{L}$. If this former loss is small, then the PDE is at least approximately satisfied.

Further, the results in Fig. E.4 seem questionable, as a standard neural network should be capable of learning this simple function. Please correct me if I am missing something important.

**Summary Of The Paper:**

The paper uses the NTK to discuss the effect of momentum on the training success of PINNs. It is shown that momentum (or adaptive momentum) can improve the convergence speed compared to standard SGD-based training, using both theoretical and empirical arguments.

**Summary Of The Review:**

An interesting paper that studies the effect of momentum on the training success of PINNs. While interesting and novel, the experimental evidence is not yet fully convincing.

---

> ### Author Response · Authors · 2022-11-18
>
> We deeply appreciate the reviewer for their insightful comments that helped us improve the quality of our paper. After carefully reading your comments and concerns, we found that there might be a fundamental misunderstanding of our paper. Specifically, **our theoretical and empirical results are consistent with what you suggested and can address your concerns regarding the experimental evidence**. In order to further clarify our results and analysis, we added Section F.2, and have modified Section 1 on page 1 and Section 3.2 on page 5 **(all color-coded in blue)** to thoroughly address your concern, regarding the validity of the spectral bias hypothesis. Below we briefly address the points: If our responses address your concerns, we hope you could raise your score.
>
> Strength And Weaknesses
> --------
> - **In fact, our (both theoretical and empirical) findings align exactly with what you suggested.** The bias of learning that is observed in our experiments and in PINNs is in general different from typical NNs: We have argued that in learning via GD optimization the overall training error over time $(t$) follows $e ^{-\kappa_i t}$, where $\kappa_{i}$ are the eigenvalues of the NTK matrix $\textit{\textbf{K}}$ that is built from $\pmb K_{bb}$ (representing the initial and boundary data points), and $\pmb K_{rr}$ (representing the PDE), with their corresponding eigenvalues $\kappa_{i}^{bb}$ and $\kappa_{i}^{rr}$. We show that for the example provided for Poisson's Equation in Section 4.2, the eigenvalues of $\pmb K_{rr}$ are much larger than the eigenvalues of $\pmb K_{bb}$ (see Section F.2 and Fig F.1). Hence, unsurprisingly, in GD where the training errors decay based on $e ^{-\kappa_{i} t}$ the network learns the PDE general form first ($\kappa_{i}^{rr}$ are larger) and becomes slow in learning the boundary and initial conditions ($\kappa_{i}^{bb}$ are much smaller). **This is the relationship with spectral bias, as it means the initial/boundary conditions are actually ``high-frequency'' components with respect to learning**. We also modified the Introduction of the paper on page 1 to explain the spectral bias phenomenon in more detail.
>
> - **Regarding the paper suggested by the reviewer (arXiv:2109.09338) in Section III.B the authors have also confirmed that in PDEs the discrepancy between PDE residuals and initial/boundary points is a general problem for learning and becomes more severe in the high-frequency target functions, just as exemplified in our Figure 5 and 6**. We agree with this, and in Section F we are just arguing that the underlying cause for this discrepancy is exactly the bias that is exhibited by GD. In Section F.2, we also showed the compound loss functions for GD and GDM cases (Fig F.2).
> - As for Figure F.3, we do not intend to show that a normal NN can or cannot learn a solution. Rather, **we wish to demonstrate that an NN learns differently from a PINN and what the PINN loss introduces, geometrically**. During their training, a regular NN will exhibit a form of spectral bias as we see the low frequencies are learned first, though the initial and boundary conditions are fairly satisfied. In a PINN, the large eigenvalues of the PDE of the NTK will cause it to learn the higher frequencies much faster. However, the inclusion of an initial/boundary condition loss introduces an NTK with small eigenvalues, which causes difficulties in learning these conditions (that a NN under mean-squared loss does not have).
>
> Clarity
> ------------
> We modified the paper accordingly.
>
> - We meant optimal solution $w*$.
> - We fixed the equation.
> - We fixed our notation to match with Eq. 1.
> - m is analogous to the mass point. We have shown in Appendix C.2 that m is related to the momentum term of GDM.
>
> Minor Comments
> ---------------
> Thanks for Catching the mistakes we had them fixed.

---

> > ### Comment · Reviewer_3LnV · 2022-11-21
> > **Thanks!**
> >
> > Thank you very much for the clarification. I indeed misunderstood a large part of the spectral bias discussion in your work, and your reply and Section F.2 clarified this to some extent. I am tending to improve my score now.
> >
> > However, there are two aspects that keep me a bit reluctant to provide a better score.
> > - First, the explanation here and in Section F.2 does not fit with Figure F.1. There, it can be seen that the eigenvalues of $K_{bb}$ are much larger than those of $K_{rr}$, hence the IC/BCs should be learned faster. I assume that this is simply because the figures are misplaced, but I would appreciate a clarifying comment.
> > - Second, the alternative hypothesis -- that the learning process is strongly affected by trivial functions, is not yet ruled out. Hence, the spectral bias is just one possible explanation for the observed phenomena. Any insights into this would be highly appreciated.

---

> > > ### Author Response · Authors · 2022-11-28
> > > **Response to the new Comment**
> > >
> > >
> > > We sincerely thank you for your comments and for the time you put into reading our response. In Fig F1, the captions were swapped. We apologize for the confusion. Caption should be: Left panel: Eigenvalues for $\pmb K_{rr}$. Right panel: Eigenvalues for $\pmb K_{bb}$.
> > >
> > > For your second concern about the validity of our hypothesis, we have pointed out the following. If our responses addressed your questions and concerns, we hope you could provide a better score:
> > >
> > > - In the case of trivial solutions of the form $u(x,w) = 0$: As mentioned in Proposition 1 of arXiv:2109.09338, this happens only in the extreme case when $n \to \infty$, which is equivalent to all the weights being initialized to 0. However, in practice, we never initialize all the weights to be 0 (see, for example, arXiv:2010.01092, where they initialized weights at random, from $N(0,1)$). Thus, it is unlikely that our estimated solutions to be trivial.
> > > Our numerical experiments show this too. Even for GD and a wide network (width 500), the estimated solution has already learned the general sinusoidal shape after just 5 epochs, rather than just learning the trivial solution of a constant value (please see the Figure in the following anonymous link <https://github.com/iclrpaper1796/Response/blob/main/5epochs.png>). We would like to also add that in practice we have shown that the PDE term of the loss function is learned much faster. This is in line with the mentioned paper (Section 3B), where they have also argued that the PDE residual is learned much faster.
> > >
> > >
> > > - The other mentioned paper, arXiv:2203.13648, is fundamentally different from our work and the work of arXiv:2109.09338. The loss function used in the paper (Equ. 2) only has the PDE residual and does not have the initial/boundary condition (IC/BC) term of a loss function in a typical PINN. Moreover, there is a difference in the terminology: by a "trivial" solution of $y = 0$, the authors of the mentioned paper meant trajectories (solution in time) towards a fixed point of $y = 0$, which is different from the "trivial" solution defined and studied for PDEs in general. Here, we explain this difference in terminology in more detail:
> > >
> > >
> > >   arXiv:2203.13648 analyzed an ODE presenting a dynamical system. In dynamical systems (when used to explain real physical phenomena) the trajectories (solutions evolving in time) are moving towards the stable fixed points and farther from the unstable fixed points. In the example, $\frac{dy}{dt} = y(y^2 -1)$ studied in the paper, the unstable fixed point is $y=0$ (here, $y$ is a fixed point and not the estimated solution of their network) and the stable fixed points are $y=\pm 1$. Later, they show if IC/BC conditions are not used, instead of the trajectories (the estimated solutions) moving toward the stable fixed points $y = \pm 1$, they will stay in the vicinity of the unstable fixed point $y = 0$. Due to the trajectories moving to an unstable fixed point, the estimation of the learned network does not necessarily have a physical meaning. The authors thus argue that IC/BC are essential as they guide the network toward true physical meanings.
> > >
> > >
> > >   We would also like to emphasize that this is not the trivial solution of $y = 0$ (in the sense of the previous paper), but rather the trajectory toward the unstable fixed point $y=0$. These ``wrong'' (unphysical) solutions are not trivial (i.e. are not constant), as seen in Fig 2 of arXiv:2203.13648.

---

> > > > ### Comment · Reviewer_3LnV · 2022-11-28
> > > > **Thanks!**
> > > >
> > > > Thank you very much for the clarification and for thus responding to my remaining concerns. I appreciate the effort, especially the additional experiment, and I agree that my claim that arXiv:2203.13648 talks about trivial solutions was wrong. In general, I have the feeling that all my concerns have now been taken care of to some extent, and I will thus improve my score.

---

### Official Review · Reviewer_droF · 2022-10-24

**Confidence:** 3
**Clarity, Quality, Novelty And Reproducibility:** See above.
**Correctness:** 3
**Technical Novelty And Significance:** 2
**Empirical Novelty And Significance:** 2
**Recommendation:** 1

**Strength And Weaknesses:**

**Weakness**

1. The biggest weakness of this paper is that the analysis does not utilize the property of physical-informed loss. In other words, the same results also hold for image classification tasks as the analysis does not rely on which loss is used. This makes the result hard to unveil any information specific to the NN-solving-PDE task.

2. There are some overclaims in the paper. It is said that "using the SGD with momentum (SGDM) optimizer can reduce
the effect of spectral bias in the networks", but the paper only proves the result for a continuous approximation of GDM with no approximation error provided.

2. The paper is hard to follow, and many places lack explanations. For example, at the end of page 3, it is said "the norm of the Hessian $\mathcal{H}(w_t)$ is order $\mathcal{O}(C^4)$". However, the parameterization of $u(w,x)$ is not even provided (although it is explained in the appendix). On page 4, it is said that "the decay rate analysis becomes more involved as Eq. 7". However, there is no Eq. 7 in the main text. Also, why does using SGDM lead to faster convergence for high-frequency components than SGD? How do we know it from Theorem 1?

**Summary Of The Paper:**

This paper studies the spectral bias phenomenon when using momentum-based methods to optimize physical-informed neural networks. The authors simplify the neural networks as a linear model (i.e., NTK), and prove that the continuous approximation of GDM converges to the high-frequency solution faster than SGD. The authors also investigate the convergence of Adam over PINN through a specific model. Experiment results are provided to show that Adam and SGDM converge faster than SGD when solving the PINNs.




**Summary Of The Review:**

Based on the evaluation above, I believe that this paper is below the bar of ICLR. The paper can be improved by refining the writing and the theoretical results.

---

> ### Author Response · Authors · 2022-11-18
>
> We thank the reviewer for the detailed comments. However, after carefully reading your review, we realize that there could be a miscommunication that leads to a fundamental misunderstanding of our paper. We hope our responses below can clarify the concerns raised.
>
> Response To Comment 1
> ---------------
> We believe this is a critical misunderstanding. In fact, Section 3.2 extensively investigates the training dynamics of PINNs under the effect of momentum. The second term of the first equation in Theorem 1 directly represents the loss in PINNs through the differential operator. Moreover, the block matrix $\textbf{K}$ uses the NTKs associated with both the PDE and boundary/initial loss functions specific to PINNs. Indeed, what makes the dynamic analysis of PINNs challenging is finding an approach which correctly implements the PINNs loss. Hence, Theorem 1 represents these training dynamics of PINNs under GDM.
>
> Response To Comment 2
> ---------------
> Thanks for pointing this out. The general steps for updating the weights in SGD and GD are similar. However, we have changed all references from SGD to GD and SGDM to GDM. Of note, previous theoretical analyses done for spectral bias were also based on GD, which is **quite common in the literature (*Cao et al. 2019* and *Wang et al. 2022*)**, and in line with our approach. However, with all due respect, we disagree with the rest of comment 2, as using a continuous approach is very common in the field and is far from a weakness of the paper (see for example works of arXiv:1902.06720v4 and arXiv:2007.14527). Besides, the extensive analysis of our numerical experiments is indicative of a very small approximation error in the estimation of the solution of PINNs via the momentum approach.
>
> Response To Comment 3
> ---------------
>
> - Due to the page limit, we have to move some details to the Appendix. As you noted, many explanations were provided in the Appendix, including the calculations of the Hessian norm and the parametrization of $u(w,x)$, which makes this paper self-contained.
> - Thanks for catching our typo. By equation 7, we meant 6, and we have corrected this.
> - We have extensively explained how and why the momentum term has resulted in faster convergence. This is essentially what our entire work is about. In fact, Section 3.2 as well as Appendices B, C, D, E, and F are all devoted to explaining, demonstrating, and iterating upon Theorem 1. On Pages 4 and 5, immediately after Theorem 1 is stated, we explain all three possible cases for the solution of equation 6 (and Figure 1 provides a visual explanation) and what they each imply for the convergence of GDM under different frequencies. As recognized by other reviewers, our theory, explanation, and analysis are in fact the main contributions and novelties of this work.
>
> References
> -----------
> 1. Yuan Cao, Zhiying Fang, Yue Wu, Ding-Xuan Zhou, and Quanquan Gu. Towards understanding the
> spectral bias of deep learning. arXiv preprint arXiv:1912.01198, 2019.
>
> 2. Sifan Wang, Xinling Yu, and Paris Perdikaris. When and why pinns fail to train: A neural tangent
> kernel perspective. Journal of Computational Physics, 449:110768, 2022.

---

> > ### Comment · Reviewer_droF · 2022-11-29
> > **My concern remains**
> >
> > I would like to thank the authors for the detailed response. I have also read the review of other reviewers and checked the paper again. However, my concerns remain.
> >
> > **Additional comment regarding Comment 1:**  I did notice that the gradient flow of GDM in Theorem 1 has a dependency on the loss in PINN. However, the rest of the analysis does not rely on the specific form of $K$. $K$ seems to be treated as a whole and the used term (e.g., the $i$-th eigenvalue of $K$) does not reflect the structure of $K$ which is composed of 4 sub-matrices. That being said, for any supervised learning task using NTK, a similar gradient flow can be obtained (with the equations for $D(u)$ removed), and the exact analysis can be carried out as we do not use the specific formulation of $K$. I expected a discussion on, for example, a specific PDE about how its $K$ behaves, and how it defers from the analysis of a supervised learning task. Otherwise, why don't we extend the analysis to general supervised learning tasks?
> >
> > **Additional comment regarding Comment 2:** I have some questions regarding the formulation of the gradient flow of GDM. First, while the authors argue that "using a continuous approach is very common in the field", I would like to argue that it is the case for GD but not for GDM. This is because you will obtain the gradient flow of GD if pushing the learning rate of GD to $0$, but you will not get the gradient flow of GDM if pushing the learning rate of GDM to $0$ with the momentum parameter $\alpha$ unchanged. $\alpha$ needs to increase to $1$ in order to ensure such an approximation (which can also be seen in Appendix C.2), and separate the analysis from the real-world applications (where $\alpha$ is usually fixed). The claim "the extensive analysis of our numerical experiments is indicative of a very small approximation error in the estimation of the solution of PINNs via the momentum approach" only shows that GDM converges but does not show that it is close to its continuous approximation.
> >
> > Second, I have some questions regarding the proof: can you explain how the second equation of equation (C.1) (dynamics of $D[u]$) is obtained? The context seems to only support the first equation of equation (C.1), which characterizes the dynamics of $u$. And what does the "solution" in "These give rise to the individual general solutions of the form" on page 14 mean? Is it the solution of $u$? If yes, why could it have no dependency on $w$? (Similar question for the "solution" in Theorem 1)
> >
> > **Additional comment regarding Comment 3:** Despite that the authors argue that "extensively explained how and why the momentum term has resulted in faster convergence", I do not understand why momentum accelerate the training from the result of Theorem 1. For example, in the underdamped case, the convergence rate of GDM is in order $e^{-\gamma t}$ while that of GD is in order $e^{-\kappa_i t}$. What can we say for this case?

---

> > > ### Author Response · Authors · 2022-12-10
> > > **Response to Comment 1**
> > >
> > > We thank you, and we appreciate the time you put to read our replies. We are sorry that our detailed response as well as the comprehensive reviews of other reviewers did not help. Here, we address your concerns:
> > >
> > > # Comment 1
> > >
> > > It is true that our theory and approach can be adapted to the typical data-driven NNs in general. **This is far from a weakness of our work, and in fact, it's the strength of the Theorem. In developing new theories, it is common to ensure that in the limit of dropping the new terms one would get the results of classic cases.** However, our work was focused on PINNs, which is why we explored Theorem 1 in the context of PINNs. **Specifically, we did analyze how $\pmb K$ behaves for specific PDEs and how this behavior differs from supervised learning**.  Here we will clarify both of these points in more detail:
> > >
> > >
> > > - $\pmb K$ contains submatrices related to the PDE and IC/BC. As $\pmb K$ is positive-semidefinite it may be diagonalized, and so there exists $N_{r}+N_{b}$ eigenvalues: the diagonal elements of the diagonalized matrix. Here, $N_r$ is the number of collocation points, and $N_{b}$ is the number of IC/BC points.
> > >
> > > Trace is defined in two (equivalent) ways: the sum of the diagonal entries of a matrix, or the sum of its eigenvalues. Let $\kappa_i^{bb}$, $i \in [1,N_b]$, and $\kappa_i^{rr}$, $i \in [1,N_r]$ be the eigenvalues of $\pmb K_{bb}$ and $\pmb K_{rr}$ respectively. Then,
> > > Then,
> > > $$
> > > \sum\limits_{i=1}^{N_r+N_b}\kappa_i= Tr(\pmb K) = Tr(\pmb K_{bb})+Tr(\pmb K_{rr}) = \sum\limits_{i=1}^{N_b}\kappa_i^{bb} + \sum\limits_{i=1}^{N_r}\kappa_i^{rr}
> > > $$
> > > **Hence, the overall convergence rate is characterized by the eigenvalues of ${\kappa_{i}}^{bb}$ and ${\kappa_{i}}^{rr}$ together.** This is how our specific choice of NTK plays a significant role in Theorem 1.
> > > Though this analysis can be performed only using the $\kappa_i$ and not considering whether they come from $\pmb K_{bb}$ or $\pmb K_{rr}$, our work in fact heavily leverages this relationship. **A comprehensive discussion related to the role of the eigenvalues of $\pmb K_{bb}$ and $\pmb K_{rr}$ is provided in Appendix F, where we analyze a specific PDE (Poisson's equation). Specifically, we explain how the discrepancy in magnitudes between the eigenvalues of the two matrices causes slow learning under GD**. We show the numerical values of the eigenvalues of the $\pmb K_{bb}$ and $\pmb K_{rr}$, and provide a plot of the training loss function for both the boundary and collocation networks (to make clear these are two networks that are trained simultaneously, and to compare how they learn in the context of spectral bias).
> > >
> > > - In terms of general supervised learning, we have already shown how PINNs differ in the learning process from a normal data-driven approach. **This is covered in Appendix F for the Poisson equation, where we show the trade-offs between the different ways a PINN and a traditional NN exhibit spectral bias. We argued how the meaning of ``high-frequency'' changes between these two networks, and how the PDE term of the loss function significantly helps enforce the general solution of a PDE.**

---

> > > > ### Author Response · Authors · 2022-12-10
> > > > **Response to Comment 2**
> > > >
> > > > Thanks for the comment, here is our response.
> > > > # Comment 2
> > > >
> > > > - In the proof provided in Appendix C.2, a continuous form of GDM was not used. Instead, a discretized form of the equation of motion of the mass point under friction (Eq. 3) was provided. **Further, it was shown that the discrete form of that equation of motion is equivalent to GDM. We respectfully disagree with the reviewer's statement the continuous approximation is only used for GD. In fact, the mentioned approach used in this work is not new and has been introduced by *Polyak 1964*,** and later expanded by *Qian 1999*. Moreover, it is not a concern that $(\alpha \to 1$) as for poorly conditioned problems it is standard practice to choose $\alpha$ very close to $1$. **This is in line with *Polyak 1964* and *Goh 2017*, where they proved that to achieve global convergence the momentum term should be chosen very close to 1. Such a choice of $\alpha$ close to \(1\) is thus not only practical but also becomes essential when the problem’s conditioning is poor (like the case of PINNs in this study)**.
> > > >
> > > > - Here, we realize that we have abused notation, which caused the confusion: **by ${D}[\ddot{u}]$ we meant $\frac{d^{2}D[u]}{dt^2}$, and by ${D}[\dot{u}]$ we meant $\frac{d D[u]}{dt}$**. We will correct our notation through the text in the paper. We would like to also add that, in the original paper on PINNs, *Raissi et al. 2017* developed two networks, one of which represented IC/BC points, and the other network represented the physics (PDE). They argued that shared parameters between the two networks can be learned by minimizing a loss function that is presented in Eq.2. **Thus, we can simply look at all PDEs ($D[u]$) with different forms as the target functions of a normal network whose inputs are the collocation points, hence each of them can separately be trained (while $\pmb K$ couples them together)**. Thus, the general dynamics of GDM that is used to explain the first equation in C.1 (evolution of $u$) is used for the second equation (evolution of $D[u]$).
> > > >
> > > > - Regarding the role of $w$ in the final solution, please note that the NTK elements are defined as:
> > > >
> > > >
> > > > $$\pmb K_{bb}(x,x') = \nabla_w{u(\textit{\textbf{w}},\textit{\textbf{x}})}^\top  \nabla_w{u(\textit{\textbf{w}},\textit{\textbf{x}}')} $$
> > > >
> > > > $$\pmb K_{br}(x,x') = \nabla_w{u(\textit{\textbf{w}},\textit{\textbf{x}})}^\top
> > > >  \nabla_w{D_u(\textit{\textbf{w}},\textit{\textbf{x}})}^\top$$
> > > >
> > > > $$\pmb K_{rr}(x,x') = \nabla_w{D_u(\textit{\textbf{w}},\textit{\textbf{x}})}^\top
> > > >  \nabla_w{D_u(\textit{\textbf{w}},\textit{\textbf{x}})}^\top,$$
> > > >
> > > >
> > > > where $D_u = D[u]$. The solution is given in terms of the eigenvalues $\kappa_i$ of the NTK, which depends on $w$, and so it does play a major role in the solution.
> > > >
> > > >
> > > > # Ref:
> > > > - Ning Qian. On the momentum term in gradient descent learning algorithms. Neural Networks, 12
> > > > (1):145–151, 1999
> > > > - Goh, Why Momentum Really Works, Distill, 2017.
> > > > - Maziar Raissi, Paris Perdikaris, and George E Karniadakis. Physics-informed neural networks: A
> > > > deep learning framework for solving forward and inverse problems involving nonlinear partial
> > > > differential equations. Journal of Computational Physics, 378:686–707, 2019
> > > > - Polyak 1964, Some methods of speeding up the convergence of iteration methods, USSR Computational Mathematics and Mathematical Physics, 1964, Pages 1-17

---

> > > > > ### Author Response · Authors · 2022-12-10
> > > > > **Response to Comment 3**
> > > > >
> > > > > Thanks for your comment here is our response:
> > > > > # Comment 3
> > > > >
> > > > > In the under-damped case, the rate of decay is not $e^{-\gamma t}$, and we never claimed such a thing. In the under-damped case, where $\gamma^2 < \kappa_{i}'$, **the radicand in Eq. 6 is negative and thus the $(\lambda_{i_{1,2}}$) are complex**:
> > > > > $$\lambda_{i_{1,2}} = -\gamma \pm i\omega_{1}$$
> > > > > where $\omega_{1} = \sqrt{\kappa_i'- \gamma^2}$.
> > > > > As a result, Eq. 6 can be re-written as:
> > > > > $$A_1 e^{(-\gamma + i \omega_{1}) t} +  A_2 e^{(-\gamma - i \omega_{1}) t}= e^{-\gamma t} (A_1e^{ i \omega_{1} t} +  A_2 e^{- i \omega_{1} t})$$
> > > > > Euler's identity states $e^{ \pm i \omega_{1} } = \cos \omega_{1} \pm i \sin \omega_{1}$, so from the above we get:
> > > > >
> > > > > $$e^{-\gamma t} [i (A_1 - A_2) \sin \omega_{1}t + (A_1 + A_2) \cos \omega_{1}t].$$
> > > > >
> > > > > Now, one can set $A_1 + A_2 = C$, and $i (A_1 - A_2) = B$, and simplify the equation as:
> > > > > $$ e^{-\gamma t} [B \sin \omega_{1}t + C \cos \omega_{1}t].$$
> > > > > With the even further simplification of $A = \sqrt{B^2 + C^2}$, and $\tan \phi = \frac{-C}{B}$, we have:
> > > > > $$ A e^{-\gamma t} \cos{(\omega_{1} t + \phi)},$$ which is given in the paper.
> > > > >
> > > > > **In this case, the rate of decay follows a sinusoidal trend (see part of equation related to $\cos(\omega_{1} t + \phi)$), and the amplitude of the sinusoidal trend decays based on $e^{-\gamma t}$. This oscillatory decay is completely different from exponential decay, and this is shown in Fig. 1 (the blue curve). Critically, the oscillatory motion forces it to be significantly slower than regular exponential growth on average.**
> > > > >
> > > > > **In the paper, we have also provided general solutions for under-damped, critically-damped, and over-damped cases**. Further details on the mathematical calculations and physical interpretations can be found in any elementary classical mechanics textbook (for example, *Arya 1990*) and are beyond the scope of this paper. We would like to repeat that unlike vanilla GD where the training error decays at the rate $e ^{-\kappa_{i} t}$, in GDM the decay rate depends on the values of the eigenvalues of the NKT, and can exhibit three different rates. These eigenvalues are related to the target function frequencies, which aids in training. We believe that we have extensively explained this point in our paper.
> > > > >
> > > > > # Reference:
> > > > > - Atam P Arya. Introduction to classical mechanics, 1998

---

### Official Review · Reviewer_Ntw5 · 2022-10-25

**Confidence:** 4
**Correctness:** 4
**Technical Novelty And Significance:** 3
**Empirical Novelty And Significance:** 4
**Recommendation:** 8

**Clarity, Quality, Novelty And Reproducibility:**

I think the paper is clearly written and results are reproducible. I don't believe anyone has used NTK to analyze PINNs for this context.

**Strength And Weaknesses:**

-The authors do a good job of introducing the reader to spectral bias in a general context in addition to spectral bias in the case of PINNs.---Very clear and concise analysis for section 3.1 and 3.2.
-Nice experimental results, they seem to confirm theoretical convergence results.


-The authors could flush out section 3.3; maybe move some of the material in the appendix for convergence results into main paper and list equations in format similar to 3.1 and 3.2
-Would be nice if authors concocted a numerically tractable ay to extract exact eigenvalues of Hessians to see how it matches with theory.


**Summary Of The Paper:**

In this paper the authors study the convergence of infinitely wide PINNs on PDEs that have high frequency modes to examine the effects of spectral bias. Their analysis proceeds using NTK theory of convergence. They show that while SGD can learn high frequency modes that the learning rate has to be small. Further they show that SGD with momentum as well as ADAM reduces the effects of spectral bias and speed up the learning. The paper also features empirical results on the poisson equation, reaction-diffusion equation and transport function.

**Summary Of The Review:**

Overall a good paper, easy to understand what the authors aimed to do. Seems like an important and useful result and ties in considering the spectrum of the physics problem in consideration with convergence guarantees of the network.

---

> ### Author Response · Authors · 2022-11-18
>
> We sincerely thank you for the positive comments and also for recognizing the contributions of our work! Here, is our response to your comments.
>
>
> - Thanks for your suggestion, we moved Section 3.3 to the Appendix, and we added Section 4.2 to our numerical analysis.
> - We have started working on developing codes to show the eigenvalues of the Hessian, we are hoping that the plots be ready before the publication of the paper.

---

### Official Review · Reviewer_3BS7 · 2022-10-25

**Confidence:** 3
**Correctness:** 4
**Technical Novelty And Significance:** 2
**Empirical Novelty And Significance:** 2
**Recommendation:** 6

**Clarity, Quality, Novelty And Reproducibility:**

# clarity

The main motivation of the paper, the existence of spectral bias, is not introduced very well. It is only mentioned that it is a known problem, but the referenced papers do not give clear definitions of spectral bias either. There should be some discussion of how to measure spectral bias, and what it is. Otherwise the paper is fairly well written, except for some typos and grammatical mistakes. The figures have bad quality and are too small.

# quality

seems to be good

# novelty

the analysis is important, but novelty is limited, there is no specific new method introduced.

# reproducibility

code is not given, but should be possible to reproduce from the paper

**Strength And Weaknesses:**

# Strengths

- for the first time, PINNs under SGDM, and Adam are analyzed, and their relation to solving spectral bias is discussed
- theoretical analysis

# Weaknesses

- it has been shown that the Adam optimizer works better than vanilla SGD in many cases (unrelated to PINNs), why would you assume that it does not work for PINNs?
- the paper aims to solve spectral bias. However, the phenomenon of spectral bias is not clearly defined or introduced in the paper
- figures are of bad quality



minor issues:
- spelling mistakes (week -> weak)

**Summary Of The Paper:**

PINNs often fail to converge. The authors propose to use neural tangent kernel to solve this issue, and propose to use the Adam optimizer. They empirically demonstrate this to work on problems with high-frequency features.

**Summary Of The Review:**

PINNs and their convergence behaviour is an understudied area where any insights can be very valuable to future research. The present work attempts to do some fundamental work to investigate PINNs. It is good to do some theoretical analysis, but novelty is limited.

---

> ### Author Response · Authors · 2022-11-18
>
> We sincerely thank the reviewer as their comments have helped us improve our paper.  Here, we replied to your comments:
>
> - It is true that momentum-based optimizers (such as Adam) are known to work better than SGD in general. The focus of our work however was not just to show that they also work better on PINNs, but to specifically explain why and how momentum results in less spectral bias than SGD, which was never studied in the literature (neither in general cases nor PINNs). This work theoretically analyzes the convergence behavior of PINNs via momentum-based algorithms, which also provides new insight into this problem.
>  - Thanks for your comment, we have modified the introduction of the paper to explain the spectral bias in more detail (highlighted in blue). The second paragraph in the revised version of the paper as well as Section F.2 are devoted to explaining the spectral bias phenomenon in NNs. Here, we briefly address the highlighting points:
> \
> Spectral bias happens when the absolute values of some of the eigenvalues of the (neural tangent kernels) NTK are large whereas some other eigenvalues are much smaller: utilizing the NTK ($\pmb K$) of infinitely-wide PINNs, *Wang et al. 2022* examined the gradient flow of these networks during training. They proved that the training error decays based on $e ^{-\kappa_{i} t}$, where $\kappa_{i}$ are the eigenvalues of the $\pmb K$, as a result, the components of the target function corresponding to the smaller eigenvalues are learned in a much slower rate. This phenomenon is known as spectral bias.
> The analysis of the spectral bias in data-driven NNs based on mean-squared-error (MSE) is much simpler. In PINNs, as the loss function has two terms (MSE and the PDE residual), the analysis becomes more challenging. This is because of the fact that $\kappa_{i}$ in PINNs are the eigenvalues of $\pmb K$ that encapsulated $\pmb K_{bb}$ (NTK for boundary and initial data points) and $\pmb K_{rr}$ (NTK for collocation points representing the PDE). Thus, the convergence rate of the training error is evaluated based on the eigenvalues of $\pmb K_{bb}$, and $\pmb K_{rr}$ together (see Theorem 1). Consequently, there might be a discrepancy between the absolute value of eigenvalues of $\pmb K_{bb}$, and $\pmb K_{rr}$, meaning that the eigenvalues of one of the two matrices be much larger, and the convergence rate of the training error for them becomes much faster. Hence, the network becomes biased to learn the components corresponding to those eigenvalues first.
>
>      In this paper, we examined the role of momentum in diminishing the effect of spectral bias. In fact, we showed that when PINNs are trained via gradient decision with momentum (GDM), the decay of the training error is analogous to a point mass undergoing a damping oscillation. Thus, the training error follows  $A_1 e^ {\lambda_{i_{1}}}t + A_2 e^ {\lambda_{i_{2}}}t$, where $\lambda_{i_{1,2}} = -\gamma \pm \sqrt{\gamma^2 - \kappa'_{i}}$. Through, the paper we proved why the GDM (based on its different training error dynamics) will diminish the spectral bias.
>   - We have replicated the figures in higher quality.
>
> References:
> 1) Sifan Wang, Xinling Yu, and Paris Perdikaris. When and why pinns fail to train: A neural tangent
> kernel perspective. Journal of Computational Physics, 449:110768, 2022

---

### Author Response · Authors · 2022-11-18
**Response to the Reviewers**

We would like to sincerely thank all reviewers for their comments and insights. We have submitted the revised version of our paper, as well as responded to the reviewers separately. The summary of changes for the paper is listed below (changes are all color-coded in blue):

- Added clarification on the spectral bias phenomenon for PINNs in the introduction of the paper.
- Added explanation on the role of NTK in the spectral bias in Appendix F.2.

---

### Decision · Program_Chairs · 2023-01-20

**Decision:**

Reject

**Justification For Why Not Higher Score:**

There are certainly several interesting aspects in this paper, but also several open questions, and to me, these open questions were so substantial that I voted for rejection.

**Justification For Why Not Lower Score:**

N/A

**Metareview: Summary, Strengths And Weaknesses:**

This paper received two borderline scores, and two other, very pronounced scores. Of the latter two, the positive review mentions three specific strengths of the paper:
(i) a good introduction to spectral (both in a general context and specifically for PINNs)
(ii) clear and concise analysis for the theory sections 3.1 and 3.2.
(iii) convincing experimental results, which seem to confirm the theory

On the other hand, the very negative review mentions three weak points:
(iv) the theory part is not specifically related to PINNs, since the results depend only on the eigenvalues of kernel matrix K. Therefore, the specific relevance of the theoretical derivation to PINNs is not obvious.
(v)  the role and interpretation of the continuous approximation of GDM is unclear, due to the dependency of a decaying learning rate and  increasing rate of parameter alpha.
(vi) some parts of the paper are hard to follow. Among other points of criticism, it was not fully clear how and why momentum could accelerate the training. As an example, the under-damped case with a convergence rate of GDM of order  exp(-gamma t) was mentioned, where the rate of GD is essentially the same.

After the rebuttal and discussion phase, I had the general impression that both the very positive and the very negative score were a bit too extreme. Therefore, I tried to largely ignore these numerical values and to focus only on the depth of the positive and negative arguments.    My conclusion is that the first positive point (i) was shared by all reviewers, whereas the second on third one were more controversial.  Concerning (iii), one of the "borderline" reviewers initially said that "some of the experimental evidence is unconvincing", although this impression seems to have changes slightly after the rebuttal. This improvement on the experimental side, however, did not change the over-all impression of this paper: the reviewer was still not overly positive in the end.

Concerning point (ii), there was quite some disagreement, which brings us to the discussion of the critical points (iv) to (vi). I think, that all three points of criticism are somewhat valid, maybe to a different degree. Concerning the over-all conceptual idea,  I think that particularly (iv) and (vi) are severe weaknesses that could not be properly addressed in the rebuttal. In my opinion, the authors' argument that the the results were still highly specific to PINNs although the theory does not differentiate between the two types of eigenvalues ("collocation" vs  "IC/BC") is neither clear nor very convincing.
The importance of point (v) is less clear to me, and might to some degree be explained by the fact that different (sub-)communities have different notions of what "is common in the field". But I would argue that in such a situation, a more in-depth discussion of this issue would be needed in the paper.
Concerning the last point (vi) about the exponential convergence rates, I had the impression that the reviewer and the authors simply talk about different things, namely the convergence to the stationary solution expressed by the real part of the eigenvalues and the transient solution, i.e. the oscillatory behavior described by the imaginary parts, and neither the reviewer nor myself were fully satisfied with the authors' rebuttal in this regard.

After finally going over all reviews and discussions again, I think that the critical points are somewhat substantial, and that there are many open questions remaining after the rebuttal. Further, also one of the "neutral" reviewers finally indicated that he/she was "not fully convinced by the correctness of the authors' hypothesis", and even the most positive reviewer did not want to champion this paper for acceptance. Therefore, I vote for rejection.